# Cryo-EM structure of hnRNPDL-2 fibrils, a functional amyloid associated with limb-girdle muscular dystrophy D3

Javier Garcia-Pardo [1], Andrea Bartolomé-Nafría [1], Antonio Chaves-Sanjuan [2,3], Marcos Gil-Garcia [1], Cristina Visentin[2,4], Martino Bolognesi [2,3], Stefano Ricagno [2,4] & Salvador Ventura [1]✉

hnRNPDL is a ribonucleoprotein (RNP) involved in transcription and RNA-processing that hosts missense mutations causing limb-girdle muscular dystrophy D3 (LGMD D3). Mammalian-specific alternative splicing (AS) renders three natural isoforms, hnRNPDL-2 being predominant in humans. We present the cryo-electron microscopy structure of full-length hnRNPDL-2 amyloid fibrils, which are stable, non-toxic, and bind nucleic acids. The high-resolution amyloid core consists of a single Gly/Tyr-rich and highly hydrophilic filament containing internal water channels. The RNA binding domains are located as a solenoidal coat around the core. The architecture and activity of hnRNPDL-2 fibrils are reminiscent of functional amyloids, our results suggesting that LGMD D3 might be a loss-of-function disease associated with impaired fibrillation. Strikingly, the fibril core matches exon 6, absent in the soluble hnRNPDL-3 isoform. This provides structural evidence for AS controlling hnRNPDL assembly by precisely including/skipping an amyloid exon, a mechanism that holds the potential to generate functional diversity in RNPs.

Human heterogeneous ribonucleoprotein D-like (hnRNPDL) belongs to a class of conserved nuclear RNA-binding proteins (RBPs) that assemble with RNA to form ribonucleoproteins (RNPs). hnRNPDL acts as a transcriptional regulator and participates in the metabolism and biogenesis of mRNA[1–4].

Three isoforms of hnRNPDL are produced by alternative splicing (AS): hnRNPDL-1, hnRNPDL-2, and hnRNPDL-3[5] (Fig. 1a and Supplementary Fig. 1). hnRNPDL-2 is a 301 residues protein and the predominant isoform in human tissues[6]. It consists of two consecutive globular RNA recognition motifs (RRM1 and RRM2), followed by a C-terminal low-complexity domain (LCD) that maps at residues ~201–285 and a nuclear localization sequence (PY-NLS) comprising residues 281-301. hnRNPDL-1 is a longer isoform of 420 amino acids containing an additional Arg-rich N-terminal LCD[5]; it is less abundant than hnRNPDL-2 and is present mainly in brain and testis[6]. hnRNPDL-3 is a shorter and minor isoform of 244 amino acids that lacks the N- and C- terminal LCDs but conserves the PY-NLS[7].

A point mutation in hnRNPDL exon 6 causes autosomal dominant limb-girdle muscular dystrophy D3 (LGMD D3)[8–10], a rare disease characterized by slowly progressive proximal muscle weakness[11,12]. This mutation changes the conserved Asp259 in the C-terminal LCD to either Asn or His. Similarly, mutation of specific Asp residues to Asn or Val in the LCDs of hnRNPA1 and hnRNPA2 are linked to amyotrophic lateral sclerosis (ALS) and multisystem proteinopathy (MSP)[13,14]. However, unlike ALS and MSP patients, where hnRNPA1 and hnRNPA2 accumulate in cytoplasmic inclusions in muscular fibers[14,15], most LGMD D3 patients do not exhibit nuclear or cytoplasmatic protein inclusions[8].

[1]Institut de Biotecnologia i de Biomedicina (IBB) and Departament de Bioquímica i Biologia Molecular, Universitat Autònoma de Barcelona, 08193 Bellaterra, Barcelona, Spain. [2]Dipartimento di Bioscienze, Università degli Studi di Milano, 20133 Milano, Italy. [3]CRC Fondazione Romeo e Enrica Invernizzi and NOLIMITS, Università degli Studi di Milano, 20133 Milano, Italy. [4]Institute of Molecular and Translational Cardiology, I.R.C.C.S. Policlinico San Donato, piazza Malan, 2, 20097 San Donato Milanese, Italy. ✉e-mail: Salvador.Ventura@uab.cat

Structures of the amyloid fibrils formed by hnRNPA1 and hnRNPA2 LCDs have been recently determined[16,17]. For both fibrils, the PY-NLS was embedded within the ordered core. This led to suggest that, under physiological conditions, binding of the import receptor karyopherin-β2 (Kapβ2) to PY-NLS[18] would impede fibrillation, whereas its exposure under pathological conditions would allow amyloid formation[16]. However, in these studies, constructs of the LCD, alone or fused to a fluorescent protein, were used to form the fibrils, and it is unknown whether the observed assemblies would match those of the natural full-length proteins' fibrils.

Here we present the cryo-electron microscopy (cryo-EM) structure of the fibrils formed by the full-length hnRNPDL-2 isoform at an overall resolution of 2.5 Å. These fibrils are stable and bind oligonucleotides, with the associated RRM domains building an exposed solenoidal coat wrapping the structured fibril. The fibrils' core is formed by a single highly hydrophilic filament encompassing LCD residues 226-276, including Asp259. Modeling suggests that disease-associated mutations at this residue may have limited impact on fibril stability. Importantly, the fibril core does not include the PY-NLS. The hnRNPDL-2 fibril core precisely matches exon 6 (residues 224-280), which is alternatively spliced (AS) in a mammalian-specific manner[19]. These AS events are frequent in RNPs, especially at their Y/G-rich LCDs[19] and they have been proposed to be a way to regulate protein function by controlling the formation of high-order assemblies[20]; our results provide structural evidence supporting this hypothesis.

Overall, we describe the structure of a full-length RNP in its fibrillar functional conformation, providing insight into the molecular bases of LGMD D3, and illustrating how AS can control RNPs assembly by including/excluding amyloidogenic exons at their LCDs.

## Results

### hnRNPDLs localize at the nucleus and exon 6 is key for their compartmentalization

Human hnRNPDL AS renders three naturally occurring transcripts (Fig. 1a and Supplementary Fig. 1). When transfected into a HeLa hnRNPDL knockout (KO) cell line (Supplementary Fig. 2), the three ectopically expressed isoforms accumulated in the nucleus (Fig. 1b and Supplementary Fig. 3), consistent with sharing a functional PY-NLS[7]. hnRNPDL-1 and hnRNPDL-2 exhibited a granulated nuclear distribution and were excluded from the nucleolar regions. In contrast, hnRNPDL-3 was homogeneously distributed in both the nucleolus and the nucleoplasm. This indicates that exon 6 (residues 224-280), shared by hnRNPDL-1 and hnRNPDL-2 and absent in hnRNPDL-3, is responsible for the differential intranuclear compartmentalization of the isoforms, delimiting regions with higher and lower protein concentration.

### hnRNPDL-2 forms stable and non-toxic amyloid fibrils

We purified monomeric hnRNPDL-1, hnRNPDL-2, and hnRNPDL-3 by size-exclusion chromatography. Upon incubation, hnRNPDL-2 forms amyloid fibrils (Supplementary Fig. 4), in a reaction that is highly sensitive to the solution ionic strength (Supplementary Fig. 5b). At pH 7.5 and low salt (50 mM NaCl), aggregation was fast, rendering large assemblies that precipitated as the reaction progressed. In contrast, at 300 mM NaCl, the reaction exhibited characteristic sigmoidal kinetics, with the formation of highly ordered individual amyloid fibrils that strongly bind Thioflavin-T (Th-T) (Figs. 1c, e). A further increase in ionic strength (i.e. 600 mM NaCl) drastically delayed and reduced hnRNPDL-2 amyloid formation (Supplementary Fig. 5b). Thus, the fibrils formed at pH 7.5 and NaCl 300 mM were selected for structural studies. Importantly, under the same conditions, monomeric hnRNPDL-1 and hnRNPDL-3 did not aggregate into Th-T positive assemblies after 24 h (Supplementary Fig. 4b), 48 h (Fig. 1c) or even after incubation for 7 days (Supplementary Fig. 6). Additionally, these two isoforms did not form amyloids upon incubation in the presence of 50 mM or 600 mM NaCl (Supplementary Fig. 5a, c). These results point to exon 6 as the

protein region responsible for amyloid formation in hnRNPDL-2. In fact, exon 6 is missing in hnRNPDL-3, and the hnRNPDL-1 N-terminal Arg-rich LCD may effectively counteract exon 6 amyloidogenic propensity, likely by diverting the assembly towards phase-separated condensates[3].

Once formed, hnRNPDL-2 fibrils are stable and insensitive to salt or temperature (Supplementary Fig. 7). Remarkably, these in vitro formed structures are devoid of toxicity for different human cell lines up to 0.4 mg/ml (Supplementary Fig. 8), which is consistent with the observation that >30% of endogenous hnRNPDL-2 accumulates into the detergent-insoluble fraction of wild-type HeLa cells (Supplementary Fig. 9).

### Cryo-EM structure determination of hnRNPDL-2 amyloid fibrils

We used cryo-EM to investigate the molecular structure of hnRNPDL-2 amyloid fibrils (Fig. 2 and Supplementary Fig. 10). Two-dimensional (2D) classification yielded one single species of twisted fibrils. About 54,500 fibril segments from 1,114 micrographs were selected for helical reconstruction in RELION 3.1[21,22], allowing us to obtain a three-dimensional (3D) density map of hnRNPDL-2 at an overall resolution of 2.5 Å (Fig. 2b, Fig. 2c and Supplementary Fig. 11). The fibril consists of a single protofilament where hnRNPDL-2 subunits stack along the fibril (Z) elongation axis. The fibril forms a left-handed helix with a full pitch of ~357 Å, a helical rise of 4.82 Å, and a helical turn of −4.86°, featuring a typical cross-β amyloid structure (Figs. 2c–f). A molecular model could be unambiguously built de novo for the fibrils core, spanning hnRNPDL-2 residues 226-276 (Figs. 2d–e) and matching strikingly the region of exon 6 (residues 224-280), which is absent in the non-amyloidogenic hnRNPDL-3 isoform (Fig. 1a). The fibril core hosts six individual β-strands (β1 to β6) connected by β-turns and loops, yielding a sort of M-shaped fold, with approximate overall footprint of 55 Å x 40 Å (Figs. 2c–e). Data collection and refinement statistics are summarized in Supplementary Table 1.

### Structural features of hnRNPDL-2 amyloid fibrils

The core of hnRNPDL-2 amyloid fibrils, as well as exon 6, map to the C-terminal Y/G-rich LCD, comprising residues ~201–285. Accordingly, the filament amino acid composition is skewed towards Gly (27%) and Tyr (25%), which comprise >50% of the residues in the structure, and neutral-polar residues, with Asn, Gln, and Ser summing to 32%. Thus, each layer of the amyloid core is essentially stabilized through intra-subunit hydrogen bonds involving buried polar side chains, main-chain peptide groups, and ordered solvent molecules (Fig. 3a). The individual protein subunits pack on each other through an extensive hydrogen-bonding network holding the in-register β-strands together, in the typical cross-β arrangement (Supplementary Fig. 12a). Additionally, the Tyr aromatic side chains build ladders of π-π stacking interactions along the fibril elongation direction, stabilizing the inter-subunit association interface. Tyr residues are approximately equally distributed between the filament interior and the surface. The core region N-end contains two additional aromatic residues, W227 and F231, which also form individual ladders of π-π stacking interactions along the fibrillar axis; surprisingly, F231 is exposed on the fibril surface, and not buried within the core. Other hydrophobic residues such as Val, Leu, and Ile are absent, whereas they are frequently observed in the structure of pathological fibrils[23]. Gly residues are dispersed throughout the sequence, with their peptide units also involved in hydrogen bonding. The main chain of each individual subunit stacked in the fibril does not lay in a plane, with a 10 Å distance between the top of β1 strand and the edge of the turn immediately after β4 (Fig. 2c). As a result, each subunit (i) not only interacts with the layer directly above (i + 1) and below (i − 1), but also with layers (i + 3) and (i − 3). This is best exemplified by the formation of a strip of hydrogen bonds involving the side chains of Q237 in layer i and Y258 in layer i-3 (Supplementary Fig. 12b-c).

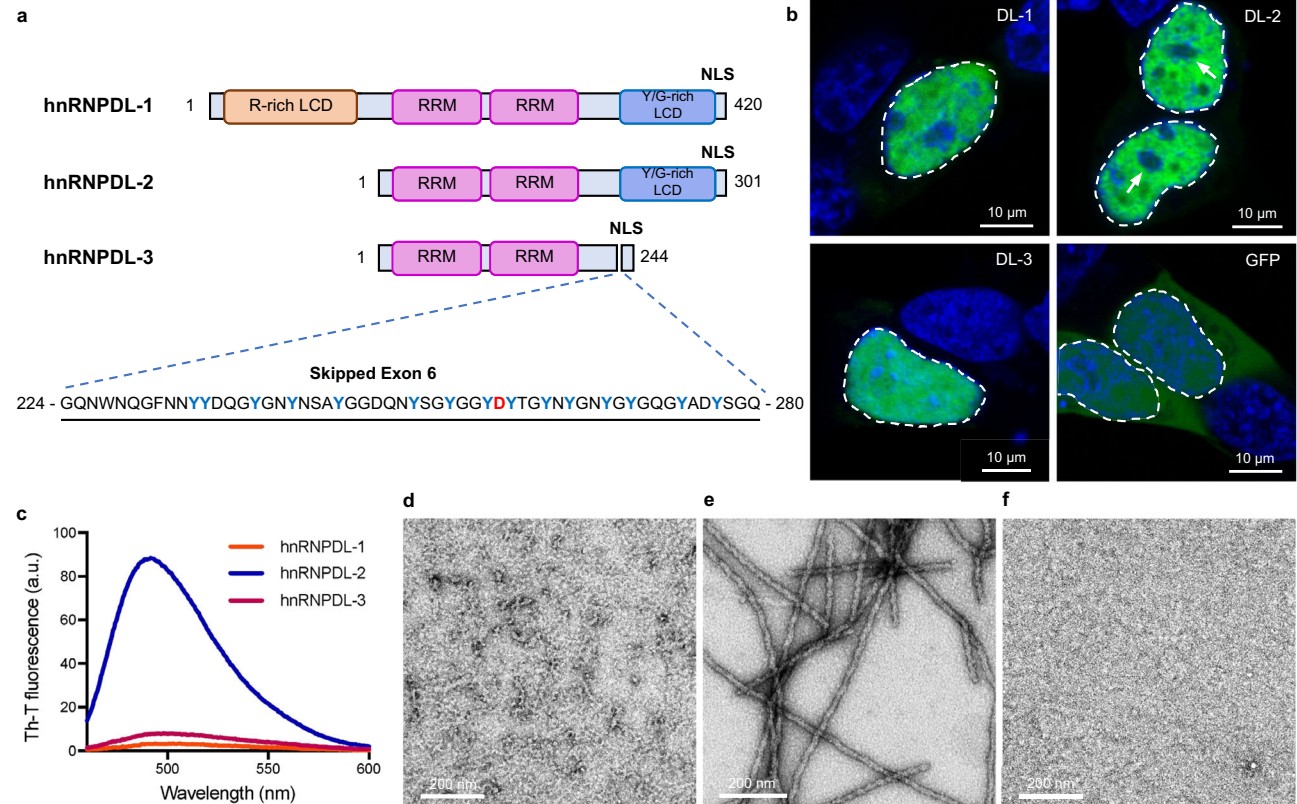

**Fig. 1 | hnRNPDL-2 aggregates into ordered amyloid fibrils. a** Domain organization of human hnRNPDL isoforms. The RNA recognition motifs are labelled as RRM (pink) and N- and C-terminal low complexity domains (LCD) are indicated as R-rich LCD (orange) and Y/G-rich LCD (blue), respectively. The region and sequence of exon 6 is also indicated, according to hnRNPDL-2 numbering. D259 disease-associated amino acid is shown in red. NLS indicates the shared nuclear localization signal. **b** Representative confocal images of HeLa hnRNPDL KO cells transiently transfected with the different hnRNPDL isoforms (DL-1, DL-2 and DL-3). The empty GFP vector (GFP) was transfected as control. The images depict the characteristic nuclear distribution of GFP-hnRNPDL fusions (green). Nuclear DNA was stained with Hoechst (blue). White arrows indicate the location of the nucleolus. In all cases, the nucleus contour has been indicated with a white dashed line. Scale bar, 10 μm. **c** Th-T binding to hnRNPDL-1 (orange line), hnRNPDL-2 (blue line) and hnRNPDL-3 (purple line) after incubation at 37 °C and 600 rpm, pH 7.5 and 300 mM NaCl for 2 days. Representative negative staining TEM micrographs of the incubated solutions of (**d**) hnRNPDL-1, (**e**) hnRNPDL-2 and (**f**) hnRNPDL-3. Scale bar, 200 nm. Note that only in (**e**) individual amyloid filaments are evident. In (**b**) and (**d**–**f**), results are representative from three independent experiments.

Unlike most amyloid structures, the conformation of hnRNPDL-2 residues N226-D276 outlines pores that define two internal water channels. The major of such channels spans the fibril's entire length and contains additional densities that correspond to two ordered water molecules per layer, that are H-bonded to nearby polar groups of Y239, N241, and with peptide N-atoms of G256 (Supplementary Fig. 12d). According to our experimental estimates, the cryo-EM density around the channel shows one of the highest resolutions within the entire fibril structure, indicating high local stability (Supplementary Fig. 11a). On average, each fibril layer encloses ten ordered water molecules.

hnRNPDL-2 fibrils are mainly composed of hydrophilic amino acids and display more polar surfaces than the fibrils of typical pathogenic proteins (Supplementary Fig. 13). The solvent-accessible surface area (SASA) of the hnRNPDL-2 fibril upper layer is 3458 Å$^2$, 48% of which is covered by polar atoms, with a percentage of polar residues in the buried area of 64%; both values being significantly higher than in disease-associated fibrils (Supplementary Table 2). The SASA of internal hnRNPDL-2 layers is, on average, 1294 Å$^2$, reflecting a burial of 63% of the surface relative to the end solvent-exposed layers, 57% of exposed atoms being polar. This endorses the lateral surface of the fibril with a high hydrophilic character (Supplementary Fig. 13a). In contrast, in the lateral surfaces of pathogenic fibrils, non-polar atoms are predominant (Supplementary Fig. 13). The percentage of polar residues in the buried area (67%) is exceedingly high when compared with the values observed for disease-associated fibrils (Supplementary Table 3).

The proportion of exposed polar atoms and polar buried residues in the inner layers of hnRNPDL-2 are also higher than in the hnRNPA1 and hnRNPA2 LCDs fibrils. Overall, hnRNPDL-2 appears to assemble into one of the most hydrophilic fibrils described so far, providing donor and acceptor groups for potential interactions.

We calculated the solvation free energy of folding (ΔG) and ΔG/residue for hnRNPDL-2 fibrils (Supplementary Fig. 14 and Supplementary Table 4). These values are significantly lower than those of disease-associated fibrils, independently if they were obtained in vitro, ex vivo or upon in vitro seeding with ex vivo fibrils, indicating that despite their irreversibility, hydrophilic hnRNPDL-2 fibrils are less stable.

Importantly, in contrast to the hnRNPA1 and hnRNPA2 LCD fibrils[16,17], the PY-NLS in hnRNPDL-2 fibrils is adjacent to, but not part of the structural core (Supplementary Fig. 15), suggesting that binding of Kapβ2 would not necessarily hamper fibrillation.

## Disease-causative hereditary mutations in the hnRNPDL-2 fibril structure

Two missense mutations in hnRNPDL exon 6, D259N and D259H, are linked to LGMD D3[8–10]. D259 is strictly conserved in the hnRNPDL C-terminal LCD of vertebrates (Fig. 3b) and maps at the end of the loop connecting β4 to β5 in the hnRNPDL-2 fibril (Fig. 3c). This residue is

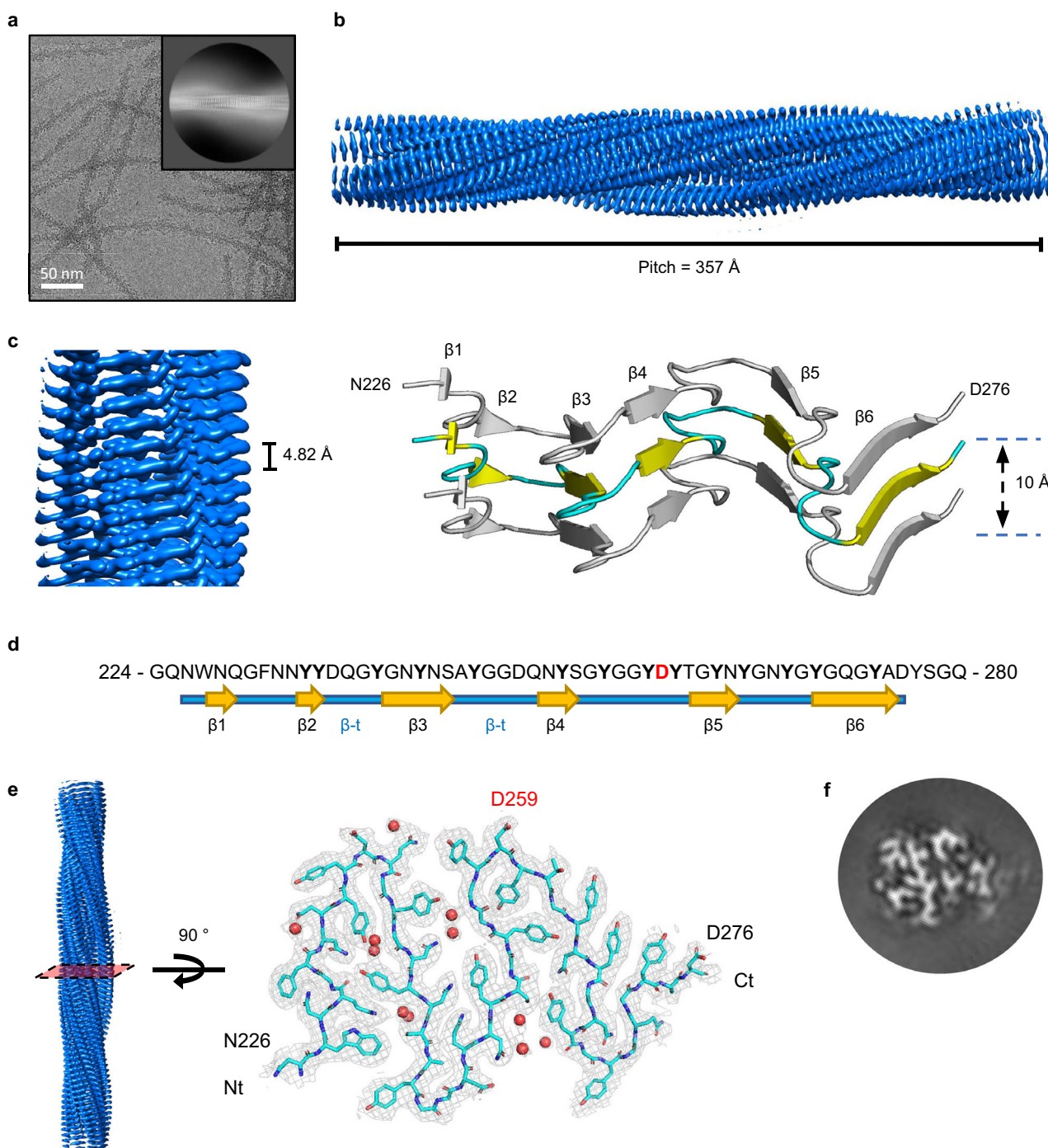

**Fig. 2 | Structure of hnRNPDL-2 amyloid filaments. a** Cryo-electron micrograph of hnRNPDL-2 filaments. Representative image from 1114 micrographs. The inset shows a representative reference-free 2D class average image of the hnRNPDL-2 filament. Scale bar, 50 nm. **b** Side view of the three-dimensional cryo-EM density map showing the structured core of an individual hnRNPDL-2 filament. The fibril pitch is indicated. **c** Detailed view of the cryo-EM density map showing the layer packing in the hnRNPDL-2 filament (left panel). The filament rise/subunit is indicated in Å. Rendered side view of the secondary structure elements accounting for three stacked rungs comprising residues N226-D276 (right panel). The distance between the stacks is indicated in Å. **d** Sequence of hnRNPDL exon 6, according to hnRNPDL-2 numbering. The observed β-strands that build the hnRNPDL-2 fibrils amyloid core are indicated. **e** Cryo-EM density map of a layer of hnRNPDL-2 amyloid core. Fitting of the atomic model for residues N226-D276 is shown on top. D259 disease-associated amino acid is highlighted in red. Water molecules are shown as red spheres. **f** Representative 3D class average image of the hnRNPDL-2 amyloid fibril.

solvent exposed, with 54% of its surface accessible in the inner fibril layers, yielding a negatively charged ladder along the fibril surface (Figs. 3c and 3d). Apart from D259, the fibril core contains three additional exposed Asp residues, D236, D249, and D276, and no positively charged amino acid, which results in a calculated single-layer pI of 3.3 and a highly anionic patch extended along the fibrillar axis.

This distinguishes hnRNPDL-2 from hnRNPA1 and hnRNPA2 LCDs fibril cores, with calculated pIs of 6.0 and 8.4, respectively.

As in hnRNPDL-2, mutation of a conserved Asp at the fibrillar core of hnRNPA2 LCD to Val (D290V) or to Asn and Val in hnRNPA1 LCD (D262N/V) are disease-associated. Virtual mutations of the respective amyloid cores[16,17] indicated that they would render more stable fibrils

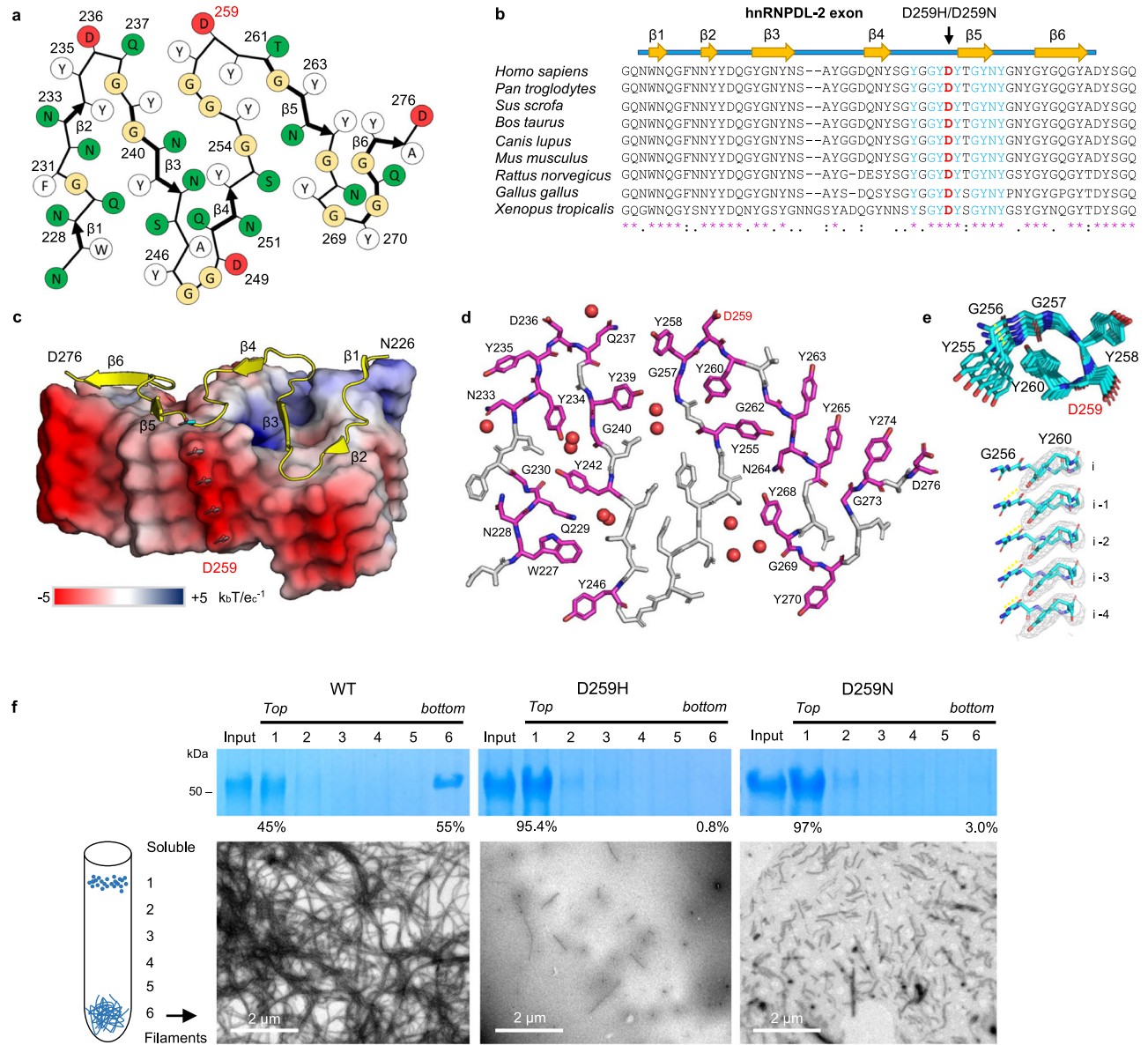

**Fig. 3 | Overall structure of the hnRNPDL-2 fibril core. a** Schematic representation of one cross-sectional layer of the hnRNPDL-2 fibril core. The location of the β-strands is indicated with thicker arrows. Polar and hydrophobic residues are colored in green and white, respectively. Glycine residues are colored in yellow. Aspartic residues are colored in red. **b** Sequence alignment of hnRNPDL orthologues showing evolutionary conservation of residues within exon 6. The β-strands that build the hnRNPDL-2 fibrils amyloid core are indicated. The mutated D259 and surrounding conserved residues are shown in red and blue, respectively. Conserved residues are indicated with an asterisk. **c** Surface representation showing the electrostatic surface potential of the hnRNPDL-2 fibril at pH 7, with ribbon representation of one subunit on top. Electrostatic potential maps were calculated using APBS server and visualized using the APBS plugin in PyMol (Schrödinger, NY, USA)[52,59]. Negative and positive map potential values are colored in red and blue, respectively, according to the $k_bTe_c^{-1}$ unit scale ($k_b$ is the Boltzmann's constant, $e_c$ is the charge of an electron and a T is 298 K). The location of the six β-strands (β1 to β6) is indicated. The side chain of D259 is shown as blue sticks and labeled. **d** Stick model of one hnRNPDL-2 fibril rung showing the position of evolutionary conserved residues (colored in pink). D259 disease-associated residue is highlighted in red. **e** Upper panel, top view of the Y255 to Y260 segment showing the interlayer interactions between Y260 and G256. Lower panel, close-up view perpendicular to the fibril axis showing the Y260 packing and their interactions with the main chain of G256. Y260 side chains are shown as sticks (colored in cyan) over the cryo-EM map sown as a grey mesh. **f** Effect of LGMDD3-associated mutations on hnRNPDL-2 fibril formation. The WT hnRNPDL-2 and D259H and D259N mutant proteins were incubated under the same conditions and subjected to separation on a glycerol cushion and SDS−PAGE analysis (fractions from top to bottom, 1–6). Bottom panels, representative NS-TEM images of amyloid filaments from the bottom fraction. This experiment has been performed twice with similar results.

by removing charge repulsions along the structure, explaining the presence of inclusions of these RNPs variants in the tissues of ALS or MSP patients[14]. However, this mechanism would not necessarily apply to hnRNPDL-2; firstly, because the mutation causing more severe and earliest LGMD D3 onset, D259H, does not neutralize the negative charge but reverts it, and secondly because the D259N mutation still leaves three other Asp residues exposed to solvent, the core surface remaining highly acidic, with a calculated pI of 3.3. Indeed, modeling

the impact of the D259H mutation in the hnRNPDL-2 fibril predicts that it would be destabilizing, whereas the stabilization predicted for D259N is lower than for the D262N/V or D290V mutations in hnRNPA1 and hnRNPA2 fibrils, respectively (Supplementary Table 5). It has been recently proposed that mutations promoting the conversion of low-complexity amyloid-like kinked segments (LARKS) into steric zippers underlie irreversible aggregation and disease[24]. The impact of the D290V mutation in hnRNPA2 LCD seems to respond to this

mechanism[24]. However, in hnRNPDL, D259H/N mutations do not increase the propensity to form steric zippers above the detection threshold (Supplementary Fig. 16). These analyses suggested that in LGMD D3, disease-associated variants may not necessarily act by increasing the fibril stability or amyloid propensity.

We produced and purified monomeric hnRNPDL-2 D259H and D259N variants and incubated them under the same conditions used to produce the wild-type (WT) fibrils described in the previous section. When the species in the incubated samples were separated on a glycerol cushion, it could be observed that the formation of high-molecular weight species was dramatically reduced in the disease-associated mutants, compared with WT (Fig. 3f). Moreover, TEM imaging of the high-molecular-weight fraction evidenced that, in contrast to the copious and long fibrils present in WT, the mutants exhibited scarce and significantly shorter fibrils (Fig. 3f), displaying poor binding to Th-T (Supplementary Fig. 17). WT hnRNPDL-2 exhibited a strict bimodal distribution, being present only in the low- and high-molecular weight fractions, indicating that they constitute the predominant metastable states of hnRNPDL-2. These results question an amyloid origin of LGMD D3 and are consistent with the absence of mutated protein inclusions in the atrophied muscle of patients[8].

### hnRNPDL-2 amyloid fibrils bind nucleic acids

hnRNPDL-2 contains two N-terminal tandem RRM domains (RRM1 and RRM2) that are thought to be functional RNA/DNA-binding motifs[25] (Fig. 1a and Supplementary Fig. 18). As shown in Figs. 4a and 4b, the RRMs were visible in hnRNPDL-2 fibrils 2D classes, when a small number of segments were averaged, as additional fuzzy globular densities around the filament core; such densities are averaged out in the 3D reconstruction due to irregular locations of the RRMs along the fibril. Importantly, the RRMs surrounding the fibrillar core could be evidenced by immunogold labeling in TEM images (Fig. 4a). We then propose a model whereby the structured fibril core, built by exon 6 residues, is decorated by a fuzzy coat of flexible RRMs (Fig. 4c).

In this respect, it should be noted that residue F231 appears exposed to solvent, giving rise to a hydrophobic ladder along the fibril hydrophilic surface. Closer inspection of the density map, however, suggests that the N-terminal arm of the fibril core, including F231, interacts with other regions of the hnRNPDL-2 assembly, likely the fuzzy RRMs (Fig. 4d), that would preclude F231 entropically disfavored interaction with the solvent.

Previous studies have shown that hnRNPDL is a protein that actively participates in transcription and AS regulation[1–4]. A prior investigation indicated that this protein binds to oligonucleotides with the consensus sequence ACUAGC, as deduced from the screening of a set of in vivo identified RNA ligands[26]. To confirm that soluble hnRNPDL-2 can bind nucleic acids, we performed Electrophoretic Mobility Shift Binding Assays (EMSA). We found that the soluble protein binds to a 7-mer fluorescently labeled ssDNA oligonucleotide displaying the ACUAGC motif with an apparent dissociation constant (Kd) of $5.9 \pm 0.78\,\mu M$ (4f, g), whereas the affinity of the protein for an equivalent RNA sequence was lower (Supplementary Fig. 19).

We incubated the same ssDNA oligonucleotide with preformed hnRNPDL-2 amyloid fibrils, and we quantified the amount of oligonucleotide bound to the fibrils after centrifugation. As shown in Fig. 4h, the fibrils significantly bind ssDNA in a concentration-dependent manner, showing a Kd of $2.1 \pm 0.36\,\mu M$. This interaction was confirmed by confocal microscopy, as incubated hnRNPDL-2 amyloids appeared highly fluorescent due to the incorporation of the fluorescein-labeled ssDNA (Fig. 4i). It could be that longer nucleic acids sequences, as those found in nature, would result in tighter binding, because multivalency allows weak interactions to collectively form much stronger interactions, a phenomenon known as avidity. Unfortunately, specific long RNA/DNA hnRNPDL targets remain to be identified. In any case, our data demonstrates that hnRNPDL-2 amyloid fibrils retain the ability to bind nucleic acids, in keeping with a potential functional role for these self-assembled structures. We have shown that the amyloid fibril surface is strongly acidic; we therefore expect the globular RRM domains that decorate the fibril to be responsible for such activity.

## Discussion

hnRNPDL-2 is the major hnRNPDL isoform in human tissues[6] and, as we show here, the only one forming ordered amyloid fibrils under physiologic-like conditions. This property can unequivocally be attributed to exon 6, which also accounts for the granular and heterogeneous protein distribution in the cell nucleus. Importantly, hnRNPDL-1 does not form fibrils, although it phase separates under the same conditions[3]. Thus, hnRNPDL constitutes a notable exception to the general rule considering liquid-liquid phase separation (LLPS) and amyloid formation as two interconnected phenomena[27,28]. This might be a way to evolve two structurally different assemblies, condensates and fibrils, each associated with an hnRNPDL isoform, skipping potential pathogenic transitions between these conformational states.

The in vitro cryo-EM hnRNPDL-2 fibrillar structure we report here exemplifies the amyloid assembly of a full-length RNP, whereas previous fibrils were solved starting from the LCD alone[16,29,30] or the LCD fused to a fluorescent protein[17,31]. Thus, we think it should better reflect the in vivo fibrillar packing of this protein family. Indeed, the hnRNPDL-2 fibril structure exhibits significant differences relative to those of hnRNPA1 and hnRNPA2 LCDs, even though the respective full-length soluble forms share the same overall molecular architecture[32,33]. In particular, the hnRNPDL-2 amyloid core matches a vertebrate conserved exon, whereas this is not the case for hnRNPA1 and hnRNPA2 (Supplementary Fig. 20). Secondly, the hnRNPDL-2 fibril is significantly more hydrophilic and acidic on its surface, a property directly impacting on the interaction with other (macro)molecules. Thirdly, hnRNPDL-2 fibrils are irreversible and cannot be disassembled at 90 °C, whereas those of hnRNPA1 and hnRNPA2 LCDs are reversible[16,17]. Indeed, ΔG and ΔG/residue values for hnRNPDL-2 fibrils are more negative than for hnRNPA1 and hnRNPA2 structures (Supplementary Table 4). In addition, our amyloid includes the complete protein, and contacts between the RRMs and the core may also contribute to stability. Indeed, reversibility of hnRNPA1 and hnRNPA2 fibrils is expected since, for them, LLPS and fibril formation are interrelated, and they can potentially transition between the two states. Such connection does not apply in the case of hnRNPDL isoforms, and because hnRNPDL-2 does not phase separate, no back-transition to liquid droplets is possible. The equilibrium should be established between the amyloid and monomeric states, which fits well with our sedimentation assays.

A final and important difference between hnRNPA1, hnRNPA2, and hnRNPDL-2 fibril structures is that, in the first two proteins, the PY-NLS residues that bind to importin Kapβ2 establish stabilizing interactions and are buried inside the fibril core[16,17]; this does not occur in hnRNPDL-2, where PY-NLS is adjacent but external to the amyloid core (not mapped by density). The accessibility of PY-NLS in both monomeric and fibrillar states of hnRNPDL-2 is consistent with the observation that, in humans, Kapβ2 co-localizes with both condensed and diffuse hnRNPDL nuclear regions[8]. This does not exclude that Kapβ2 might still modulate hnRNPDL-2 fibril formation, sterically interfering with the building of the amyloid core; however, this action would be mechanistically different from those exerted on hnRNPA1 and hnRNPA2, where Kapβ2 also acts as disaggregase[34].

hnRNPDL-2 fibrils share similarities with the 3D structures of functional amyloids. Like most of them[35–37], hnRNPDL-2 fibrils do not exhibit polymorphism. This suggests that they may represent a global free energy minimum, allowing us to speculate that the same structure would be adopted in the cell and that it represents the functional conformation of the assembled state. In contrast, pathogenic fibrils are mostly polymorphic[23]. The hnRNPDL-2 fibril core is hydrophilic, and the structure is stabilized by hydrogen-bonding networks, between

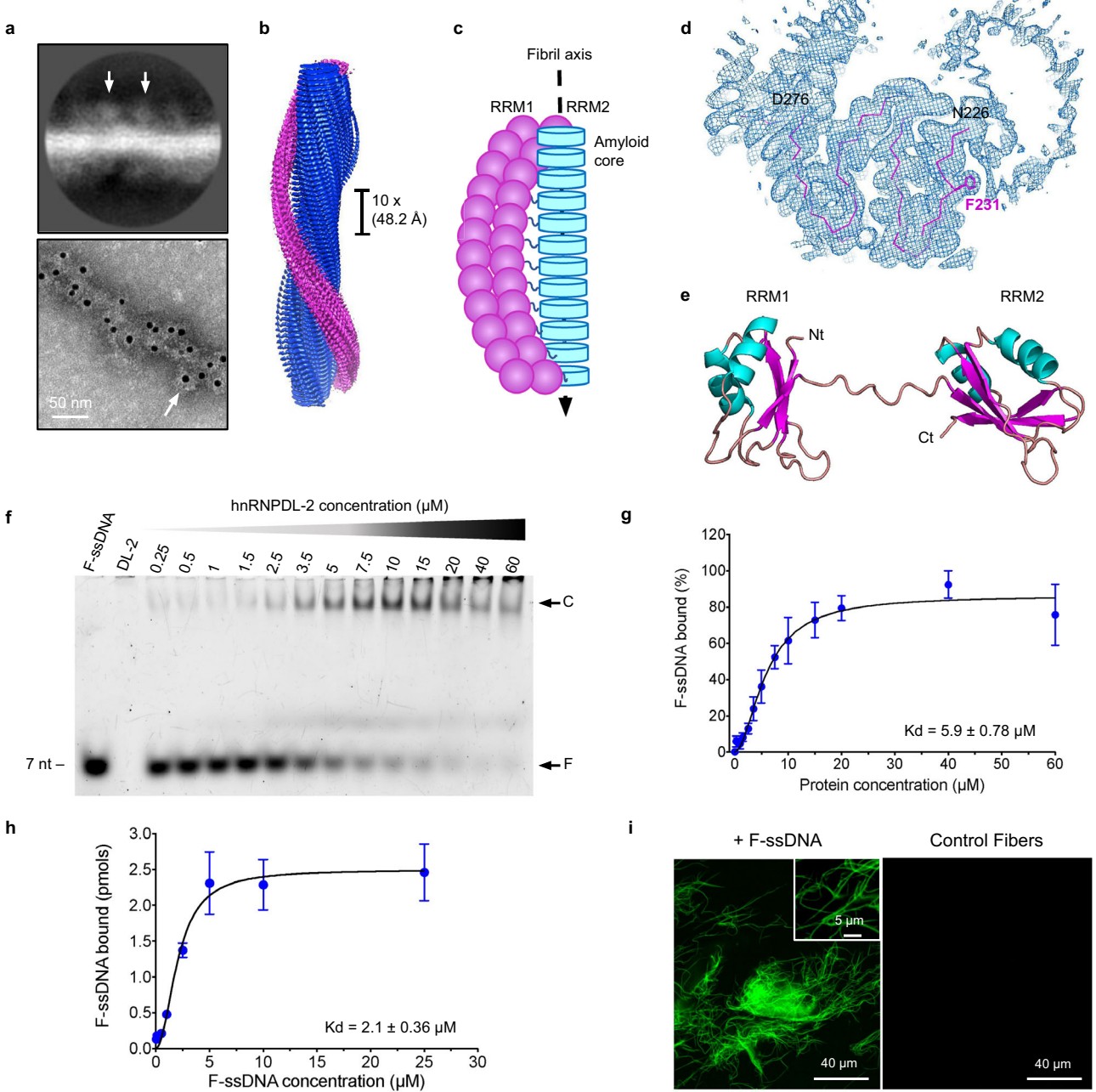

**Fig. 4 | Structure of the RNA/DNA-binding domains from hnRNPDL-2 and their location in the amyloid filaments. a** Top panel, representative 2D class average image showing hnRNPDL-2 fibrils with an approximate solenoid coat of globular domains. The arrows indicate the location of the density assigned to the RRMs. Bottom panel, negative-stain electron microscopy (EM) micrographs of hnRNPDL-2 filaments bound to nanogold-antibodies (~10 nm, white arrow) targeting the location of the hnRNPDL-2 globular domains. The N-terminal 6xHis tag was labelled using nanogold-conjugated secondary antibodies against anti-6xHis antibodies produced in mouse. Representative image from two independent experiments. Scale bar, 50 nm. **b** 3D unsharpened density map reconstruction of hnRNPDL-2 fibril. Densities for the amyloid core and putative RRMs are colored in blue and pink, respectively. The 10x filament rise/subunit is indicated in Å. **c** Schematic diagram showing the proposed organization of the RRM domains (pink) around the fibril core (cyan). **d** Cross-sectional view of the unsharpened cryo-EM map of hnRNPDL-2 fibril, with the superimposed hnRNPDL-2 amyloid core in a ribbon representation. The side chain of F231 is represented as sticks. **e** Model structure of

the N-terminal RNA-binding domains RRM1 and RRM2 from hnRNPDL-2 generated with AlphaFold2[60]. **f** Electrophoretic mobility shift assay (EMSA) of soluble hnRNPDL-2 with a Fluorescein-labelled oligonucleotide (F-ssDNA). The 7-mer ssDNA was incubated with the soluble form of hnRNPDL-2 at the indicated protein concentrations. EMSA has been performed three times with similar results. **g** Binding affinity of soluble hnRNPDL-2 to the 7-mer fluorescent ssDNA (F-ssDNA) determined by the EMSA assay. Data is shown as mean ± SEM ($n = 3$ independent experiments). **h** Binding of the 7-mer fluorescent ssDNA (F-ssDNA) to preformed hnRNPDL-2 amyloid filaments. Data is shown as mean ± SEM ($n = 4$ independent experiments for all protein concentrations, except for 10 and 25 μM with $n = 3$ independent experiments). **i** Representative confocal microscopy image of hnRNPDL-2 amyloid fibrils bound to fluorescent ssDNA (+F-ssDNA). Control fibrils without F-ssDNA are shown as control condition. Representative image from three independent experiments. In (**g**) and (**h**) data was fitted to one-site specific binding mechanism with Hill slope using GraphPad Prism[57].

residues and with water molecules, together with Tyr side chains π-π stacking and dipole-dipole interactions. The eminent polar nature of the interactions that hold up the hnRNPDL-2 fibril and the abundance of flexible Gly residues would allow protein chains to explore the conformational space efficiently towards the final structure, without being trapped into polymorphic local minima by stable hydrophobic interactions, as it often occurs for disease-associated fibrils[23]. In addition, the hydrophilic nature of hnRNPDL-2 fibril surfaces may lay behind their observed lack of toxicity, precluding interactions with hydrophobic cellular membranes and their subsequent disruption.

hnRNPDL-2 fibrils consist of a single filament, whereas 74% of the available amyloid structures have two or more protofilaments[38]. This is not surprising, since most fibril interfaces involve hydrophobic interactions between individual protofilaments[38] and, as said, non-polar amino acids are virtually absent in the hnRNPDL-2 fibril core. In addition, the surrounding RRMs coat would impair any filament-to-filament lateral association.

The structure of hnRNPDL-2 fibrils is reminiscent of that of HET-s prion, where globular domains also hang from a single filament fibril in a solenoidal fashion[35]. A single filament allows for the decoration of amyloid cores with regularly spaced globular domains following the helical twist of the fibril, something hardly compatible with a multi-protofilament assembly. Indeed, the fibrils of hnRNPA2[17] and FUS[31] LCDs fused to fluorescent proteins also exhibit a single protofilament, consistent with this being the preferred disposition when globular domains are adjacent to LCDs in the sequence. Instead, the fibrils of hnRNPA1 LCD alone involved two filaments[16].

The most substantial evidence of hnRNPDL-2 fibrils' functionality is their ability to bind small oligonucleotides, especially ssDNA, with an affinity equivalent to that of the soluble counterpart. This indicates that the RRM domains wrapping around the structured fibrils are folded and functional.

The functionality of hnRNPDL-2 fibrils provides the basis for understanding the connection between D259 mutations and LGMDD3 and why, in contrast to MSP patients bearing similar mutations in hnRNPA1 and hnRNPA2 LCDs, whose muscle biopsies show cytoplasmic mislocalization and protein aggregation[15], in LGMDD3, myopathologic studies coincide in the absence of sarcoplasmic protein aggregates[8], although congophylic deposits were detected in some instances[39]. For hnRNPA1 and hnRNPA2, theoretical calculations indicated that Asp substitutions would stabilize the fibrils[16,17] and facilitate LARKS to steric zippers transitions[24]. This would thermodynamically shift any potential droplet/fibril equilibrium towards the fibrillar state, reducing reversibility. For hnRNPDL-2, this equilibrium does not apply, and the WT fibrils are already irreversible. The same calculations indicate that D259H/N mutations do not significantly stabilize the fibril core and that no LARKS to steric zipper transition occurs. Indeed, the mutant proteins exhibit a low propensity to fibrillate, and the fibrils are shorter and less organized. This is consistent with the lack of aggregates in most patients' muscular tissues[8], suggesting that we might face a loss-of-function disease. It could be that hnRNPDL-2 fibrils cannot be efficiently formed in the affected muscle, and the soluble protein cannot compensate for their activity, or that Kapβ2 cannot properly transport the mutant proteins[7]. Alternatively, inefficient fibrillation pathways of mutants might involve intermediates that are either degraded by the protein quality control machinery, decreasing the pool of active protein, or instead accumulate in myocytes, exerting toxicity. Indeed, knockdown of zebrafish hnRNPDL (85% identity with human hnRNPDL-2) using antisense oligonucleotides resulted in dose-dependent disorganization of myofibers, causing body shape defects and restricted and uncoordinated movements, which is consistent with a loss-of-function myopathy[8].

Our results are in keeping with the recent evidence that even for hnRNPA1, disease manifestation is not always associated with increased fibrillation, and variants with LCD mutations displaying a low ability to form fibrils are also pathogenic[40]. Interestingly, these hnRNPA1 variants cause vacuolar rimmed myopathy, histologically similar to LGMDD3. This suggests that RNP-associated diseases might not respond to a unique molecular mechanism, but somewhat different sequential/structural perturbations might elicit cellular dysfunction and degeneration, potentially by affecting the function of LCD-containing RNPs in muscle regeneration, where they execute pre-mRNA splicing, stabilize large muscle-specific transcripts and aid in their transport[41].

The most intriguing and unique feature of the hnRNPDL-2 fibril is the perfect overlap between the amyloid core and exon 6. AS patterns have diverged rapidly during evolution, with exons that were ancestrally constitutive in vertebrates evolving to become alternatively spliced in mammals, expanding the regulatory complexity of this lineage[42]. These evolutionary changes impact all members of the hnRNPD family, to which hnRNPDL belongs. Accordingly, the exon 6 sequence is conserved among vertebrates (Fig. 3b), an additional evidence of its functionality, and alternatively spliced in mammals[7,19]. Mammalian-specific AS is especially frequent at the Y/G-rich LCDs of RNPs[19], suggesting that regulation of the number of GY motifs in these regions confers fitness benefit[20]. Furthermore, elimination of these repeats through exon skipping results in dominant-negative RNPs that bind nucleic acids but cannot form multimeric complexes through Tyr-dependent interactions, which significantly modifies their gene regulation activity and nuclear patterning[20]. This differential behavior is often attributed to the longer isoforms' ability to undergo LLPS[20]. However, in proteins of the Rbfox family, the splicing activity is contingent on the formation of Th-T positive fibrous structures, which is mediated by its LCD in a Tyr-dependent manner[43], although it is unknown if Rbfox fibers correspond to cross-β amyloids. Here we provide high-resolution structural evidence supporting the role of mammalian specific-AS in controlling the assembly and nuclear distribution of RNPs, which in the particular case of hnRNPDL occurs by precisely including/skipping a conserved amyloid-prone exon.

Overall, this work presents the detailed cryo-EM structure of a full-length RNP in its fibrillar functional conformation. The structure of hnRNPDL-2 exon 6 in its amyloid form provides critical insights into the molecular bases of LGMD D3 and the mechanism of AS-controlled RNPs assembly in mammals.

## Methods

### Cell culture, plasmids and cell lines
The human HeLa (ATCC CCL-2) and SH-SY5Y (ATCC CRL-2266) cell lines were maintained in Dulbecco's Modified Eagle Medium (DMEM) or minimum essential medium α (MEM-α) medium, respectively. Media were supplemented with 10% (v/v) Fetal Bovine Serum (FBS). Both cell lines were grown under a highly humidified atmosphere of 95% air with 5% $CO_2$ at 37 °C. The HeLa hnRNPDL KO cell line was generated as described elsewhere[3]. HeLa cells have been authenticated within the last 3 years by STR analysis. The HeLa and SH-SY5Y cell lines were regularly tested for the presence of mycoplasma using qPCR Detection Kit (SIGMA), and both cell lines are mycoplasma negative.

For recombinant protein expression, the genes encoding the three isoforms of hnRNPDL (hnRNPDL-1, hnRNPDL-2 and hnRNPDL-3) were inserted into pETite (Lucigen corporation) vector with a His-SUMO N-terminal tag. For subcellular localization experiments, the same genes were cloned into pEGFP-C3 (Clontech). In all the cases, the correctness of the DNA sequence was verified by sequencing.

### Cell transfection and immunoblotting
Adherent cells were transfected with linear polyethylenimine (PEI; Polysciences) in a 1:3 DNA:PEI ratio. Cells were collected after 48 h, lysed in M-PER mammalian protein extraction reagent (Thermo Fisher Scientific) with 1/1000 of the EDTA-free protease inhibitor cocktail Set III (Calbiochem), and centrifuged for 30 min at

15,000 × $g$ at 4 °C. The soluble and insoluble fractions were analyzed by SDS−PAGE and immunoblotting onto PVDF membranes (EMD Millipore) using standard protocols[44]. hnRNPDL proteins were detected using different primary antibodies (anti-hnRNPDL antibody, HPA056820 from Sigma-Aldrich, dilution 1:500; anti-vinculin monoclonal antibody VLN01, MA5-11690 from Invitrogen, dilution 1:5000). The primary antibodies were detected with the appropiate HRP-labelled secondary antibody (goat anti-Rabbit IgG (H + L), 31460 from Invitrogen, dilution 1:2000 or goat anti-Mouse IgG (H + L), 31430 from Invitrogen, dilution 1:2000), followed by chemiluminescence detection using Immobilon Forte Western HRP substrate (Sigma-Aldrich).

## Protein expression and purification
Protein expression was performed in *E. coli* BL21(DE3) cells, induced with 0.5 mM IPTG at a $OD_{600} = 0.5$. After incubation for 3 h at 37 °C and 250 rpm, the cells were harvested by centrifugation for 15 min at 4000 × $g$. Cell pellets were resuspended in Binding Buffer (50 mM HEPES, 1 M NaCl, 5% glycerol and 20 mM imidazole, pH 7.5), lysed by sonication, and centrifuged at 30,000 × $g$ for 30 min at 4 °C. The supernatant was filtered through a 0.45 μm filter and loaded into a HisTrap™ FF Ni-column equilibrated with Binding Buffer. The bound protein was eluted with an imidazole gradient starting from 0 to 100% of the Elution Buffer (50 mM HEPES, 1 M NaCl, 5% glycerol, 500 mM imidazole, pH 7.5). Afterwards, fractions containing purified proteins were pooled and loaded into a HiLoad™ 26/600 Superdex™ 75 pg column equilibrated with a 50 mM HEPES pH 7.5 buffer containing 1 M NaCl and 5% glycerol. Finally, the proteins were concentrated using a 10 K Amicon (Merck-Millipore), flash-frozen in liquid nitrogen and stored at −80 °C until use.

## Fibril formation and in vitro aggregation kinetics
For aggregation experiments, the purified proteins corresponding to the different hnRNPDL isoforms and mutants were loaded into a PD-10 desalting column Sephadex™ G-25 M to exchange the buffer. Samples were diluted to a final protein concentration of 50 μM in 50 mM HEPES, 300 mM NaCl, pH 7.5. Then, the aggregation reactions were incubated at 37 °C with 600 rpm agitation in sealed Eppendorfs.

Th-T binding to hnRNPDL aggregates was measured by recording Th-T fluorescence in the range between 460 and 600 nm after excitation with wavelength of 445 nm using a Jasco FP-8200 spectrofluorimeter. The final Th-T and protein concentrations were 25 μM and 10 μM, respectively. All the samples were diluted in 50 mM HEPES buffer at pH 7.5 with 300 mM NaCl and this same buffer alone was used as a control. The light scattering of the reactions was measured in the range between 325 and 340 nm after excitation at 340 nm using a Jasco FP-8200 spectrofluorimeter.

The aggregation kinetics of hnRNPDL isoforms were monitored in 96 well plates by following increments in Th-T signal of 25 μM protein samples. Plates were incubated at 37 °C under constant shaking (100 rpm) using a Spark (TECAN) Fluorescence microplate reader. The Th-T fluorescence of each well was measured every 30 min by exciting with a 445 nm filter and collecting the emission with a 480−510 nm filter. Fluorescence emission of all the proteins in the absence of Th-T and the signal of the buffers alone with Th-T were determined as control conditions. To test the effect of NaCl on the aggregation kinetics of hnRNPDL, aggregation reactions with different NaCl concentrations (50, 300 and 600 mM) were also prepared.

## Ultracentrifugation
To perform ultracentrifugation experiments, the samples were centrifuged at 120,000 × $g$ for 2 h at 4 °C in a 20% (v/v) glycerol cushion. After centrifugation, six fractions (70 μl each) were collected from top to bottom, without disturbing the layers, and analyzed by SDS−PAGE. The total protein concentration of each fraction was determined by Bradford using a commercial Coomassie Protein assay reagent (Thermo Fisher Scientific).

## Immunolabelling and negative staining electron microscopy
Aliquots of hnRNPDL-2 amyloid fibrils were mixed with an anti-6xHis (MA1-21315 from Thermo Fisher Scientific) primary antibody (1:10 dilution) in 50 mM HEPES, 150 mM NaCl, pH 7.5 buffer and the sample was then incubated for 18 h at 4 °C. After incubation, an anti-mouse 10 nm colloidal gold-linked secondary antibody (A-31561 from Thermo Fisher Scientific) was added to the samples (dilution 1:100) and incubated for 1 h at room temperature. For negative staining transmission electron microscopy (NS-TEM), 5 μl of each sample was incubated on a EMR 400 mesh carbon-coated copper grids (Micro to Nano Innovative Microscopy Supplies) for 5 min. After incubation, the grids were washed with 10 μl MQ water and stained for 1 min with 5 μl of 2% (w/v) uranyl acetate. The excess solutions from each step were removed with filter paper. Each grid was allowed to dry before inspection using a JEOL JEM-1400 Electron Microscope operating at 120 kV with a CCD GATAN 794 MSC 600HP camera with Digital Micrograph 1.8 (GATAN) software. 3−5 micrographs were recorded for each sample at two different nominal magnifications (*i.e.* ×6000 and ×10000) and an estimated defocus of about ±1−4 μm.

## Cryo-EM sample preparation and data collection
For cryo-EM, sample vitrification was carried out using a Mark IV Vitrobot (Thermo Fisher Scientific). 3 μl hnRNPDL-2 amyloid fibrils diluted in MQ water at a final concentration of 0.25 mg/mL were applied to a C-Flat 1.2/1.3-3Cu-T50 grid (Protochips) previously glow-discharged at 30 mA for 30 s in a GloQube (Quorum Technologies). Sample was incubated on grid for 60 s at 4 °C and 100% humidity, blotted and plunge-frozen into liquid ethane. Vitrified samples were transferred to a Talos Arctica transmission electron microscope (Thermo Fisher Scientific) operated at 200 kV and equipped with a Falcon 3 direct electron detector (Thermo Fisher Scientific) and EPU 2.8 (Thermo Fisher Scientific) software. A total of 1114 movies were collected using EPU 2.8 (Thermo Fisher Scientific) in electron counting mode with an applied dose of 40 e⁻/Å² divided in 40 frames at a magnification of 120 kx. All the micrographs were acquired with a pixel size of 0.889 Å/pixel and a defocus range of −1.0 to −2.2 μm.

## Helical reconstruction
Best movies (without the presence of artifacts, crystalline ice, severe astigmatism or with obvious drift) were imported in RELION 3.1 for further processing following a helical reconstruction pipeline[22]. All the movies were motion-corrected and dose-weighted using MOTIONCOR2[45]. Contrast transfer function (CTF) estimation was performed on aligned, unweighted sum power spectra every 4 e/Å² using CTFFIND4[46]. Micrographs with a resolution estimate of 5 Å or better were selected for further analysis. Fibrils were manually picked, and the segments were successively extracted using a box size of 380 pixel and inter-box distance of 33.2 Å, yielding a total of 158,493 segments. Reference-free 2D classification was performed to identify homogeneous segments for further processing.

An initial model was generated using a single large class average similarly as described elsewhere[47]. In brief, the cross-over distance was measured from the class average image to be ~179 Å. The initial model was then re-scaled and re-windowed to match the unbinned particles with a 380-pixel box and low-pass filtered to 10 Å. A 3D classification with four classes, a regularization value of $T = 4$ and imposing a helical rise of 4.82 Å and a helical twist of −4.86° was used to select the best class containing 54490 particles.

Particles were aligned with a 3D auto-refine job using a 10 Å low-pass filtered map from the previous 3D classification, considering the 30% central part of the box, and refining the helical twist and rise. Bayesian polishing[48,49] and CTF refinement[48] were performed to

increase the resolution of the final reconstruction. Finally, the refined 3D reconstructions were sharpened using the standard post-processing procedures in RELION 3.1. Based on the gold-standard Fourier shell correlation=0.143 criteria, the best map for the hnRNPDL-2 fibril showed a final resolution of 2.5 Å. All the details of data acquisition and processing are given in Supplementary Table S1. The hnRNPDL-2 structure, and cryo-EM map have been deposited in the Protein Data Bank (PDB ID: 7ZIR) and Electron Microscopy Data Bank (EMDB-14738), respectively. Raw images and raw processing data are available at the Electron Microscopy Public Image Archive - EMBL-EBI (EMPIAR-11064).

## Model building and refinement

The hnRNPDL amyloid fibril model was manually built into the 2.5 Å sharpened map of the hnRNPDL-2 fibril using COOT[50]. In brief, an initial polyalanine model was incorporated into the 2.5 Å sharpened map of the hnRNPDL-2 filament. Then, Ala residues were mutated to Tyr where the electron density was clear enough. Based on the location of Tyr and spacing pattern, the comprising residues 226 to 276 of hnRNPDL-2 were successfully assigned and modeled. Real-space refinement was accomplished with the phenix.real_space_refine module from Phenix[51]. Molecular graphics and structural analyses were performed with Pymol[52] and ChimeraX[53]. Statistics for the final model are provided in Supplementary Table 1 and the corresponding atomic coordinates have been deposited in the Protein Data Bank (PDB accession number: 7ZIR).

## Fibril stability and solvation free energy calculations

The free energy of mutation calculations were determined using the module ddg_monomer from Rosetta[54]. Delta delta G ($\Delta\Delta G$) energy values for WT hnRNPDL-2 and mutants (D259H, D259N and D259V) fibril structures were determined following a standard high-resolution protocol as described elsewhere[16]. In brief, we first performed energy minimization on the hnRNPDL-2 fibril structure containing five successive rungs. The resultant restraint file from this step was used in the subsequent processing. Standard Van der Waals and solvation energy parameters (i.e. cut off value of 9 Å) were applied. For comparative purposes, the $\Delta\Delta G$ values for WT and relevant hnRNPA1 or hnRNPA2 mutant fibrils were also determined using the same Rosetta protocol.

The percentage of buried polar residues and the exposed surfaces of hnRNPDL-2, hnRNPA2, hnRNPA1, hSAA1, A$\beta$−42 and $\alpha$-synuclein fibrillar structures were calculated using PDBePISA[55]. Exposed surfaces were calculated as the difference between the total and the buried surface. The percentage of exposed polar surface was calculated using the total and apolar surface values predicted by GetArea[56]. The solvation free energy of folding ($\Delta G$ in kcal/mol) values for all the structures were calculated using PDBePISA[55].

## In vivo cell imaging

For in vivo cell imaging experiments, adherent HeLa cells were grown on 35 mm glass-bottom culture dishes at a density of $1 \times 10^5$ cells per dish. After 24 h incubation, the cells were transiently transfected with GFP-tagged versions of the hnRNPDL-1/-2/-3 isoforms and linear polyethylenimine (PEI; Polysciences, Eppelheim, Germany) in a 1:3 DNA:PEI ratio. After 24 h incubation, the cells were washed with fresh medium ($2 \times 1$ mL), and then cell nuclei were stained with Hoechst (Invitrogen) for 10 min at 37 °C and 5% $CO_2$. In vivo cell imaging was performed at 37 °C using a Leica TCS SP5 confocal microscope and a $63 \times 1.4$ numerical aperture Plan Apochromat oil-immersion lens. All the confocal images were processed using Bitplane Imaris 7.2.1 software.

## Gel Electrophoretic Mobility Shift Binding Assays (EMSA)

The ability of hnRNPDL-2 amyloid fibrils to bind RNA/ssDNA was studied by performing a Gel Electrophoretic Mobility Shift Binding Assays (EMSA). We prepared 10 μl reactions containing 50 nM of Fluorescein-

labeled RNA/ssDNA (F-GACUAGC) and increasing amounts of soluble hnRNPDL-2. The samples were incubated for 24 h at RT before the complexes were resolved on an 8% polyacrylamide gel at a constant voltage. After electrophoresis, the gel was imaged using a ChemiDoc™ MP Imaging System (Bio-Rad) and Image Lab Touch Software (Bio-Rad Laboratories). Each experiment was performed in triplicate. The apparent dissociation constants (Kd) for specific binding of RNA or ssDNA to soluble hnRNPDL-2 were determined by fitting the data to one-site specific binding mechanism with Hill slope using GraphPad Prism[57].

## Fluorescent RNA/DNA-binding assays

The ability of hnRNPDL-2 amyloid fibrils to bind ssDNA was studied by measuring the fluorescence of the fibrils bound to Fluorescein-labelled ssDNA using a Jasco FP-8200 Spectrofluorometer (Jasco Corporation) and a fluorescence microscope (Leica Microsystems).

For the fluorometric assays, 20 l reactions containing 10 μM of hnRNPDL-2 amyloid fibrils and increasing amounts of Fluorescein-labeled ssDNA (F-GACUAGC) at the following concentrations (0.05, 0.1, 0.5, 1, 2.5, 5, 10, 25, 50 and 100 μM) were prepared. After incubation for 24 h at RT, the samples were centrifuged at $16,000 \times g$ for 40 min at 4 °C. The pellet containing the hnRNPDL-2 amyloid fibrils bound to the ssDNA was washed with 50 μl of 50 mM HEPES buffer (pH 7,5) with 150 mM NaCl, centrifuged for 30 min and sonicated. Fluorescence emission of the ssDNA bound to fibrils was measured at an excitation and emission wavelength of 480 nm and 520 nm, respectively. Control samples with only fluorescent ssDNA were used as controls conditions. Each experiment was performed at least in triplicate. The apparent dissociation constant (Kd) for specific binding of ssDNA to hnRNPDL-2 amyloid fibrils was determined by fitting the data to one-site specific binding mechanism with Hill slope using GraphPad Prism[57].

For confocal microscopy analysis, 20 μl reactions with hnRNPDL-2 amyloid fibrils at a concentration of 20 μM were incubated for 24 h at RT in the presence of 30 μM of Fluorescein-labelled ssDNA (F-GACUAGC). After centrifugation at $16,000 \times g$ for 40 min, the precipitated fraction was placed on a microscope slide and sealed. Confocal fluorescence images were obtained with Leica SP5 confocal microscope (Leica microsystems). hnRNPDL-2 amyloid fibrils and labeled ssDNA alone were imaged as control samples.

## Cytotoxicity assays

The cytotoxicity of the hnRNPDL-2 amyloid fibers toward SH-SY5Y and HeLa cells was evaluated using a resazurin-based assay[58]. Briefly, the cells were seeded in 96-well plates at a concentration of $3 \times 10^3$ cells per well and incubated for 24 h. Afterwards, the cells were treated with different hnRNPDL-2 fiber concentrations ranging from 0.004 to 0.4 mg/mL (w/v). After 48 h of incubation, aliquots of 10 μL of the PrestoBlue™ (Thermo Fisher Scientific) cell viability reagent solution were added to each well. After 1 h incubation at 37 °C in the absence of light and in the presence of a highly humidified atmosphere of 95% air with 5% $CO_2$, fluorescence emission (ex/em:531/572 nm) of each well was measured using a fluorescence microplate reader (PerkinElmer Victor 3 V) and PerkinElmer 2030 software. Cell cytotoxicity was determined in terms of cell growth inhibition in treated samples and expressed as percentage of the control condition.

## Reporting summary

Further information on research design is available in the Nature Portfolio Reporting Summary linked to this article.

## Data availability

The hnRNPDL-2 amyloid fibril structure, and cryo-EM map have been deposited in the Protein Data Bank and Electron Microscopy Data Bank under the accession codes 7ZIR and EMDB-14738, respectively. Raw images and raw processing data have been deposited in Electron

Microscopy Public Image Archive with accession code EMPIAR-11064. Other structures referenced in this work are available under the PDB accession codes 6MST, 5KK3, 6OSJ, 7Q4M and 7NCK. Other data supporting the findings of this study are available within the article and its associated supplementary information files. Source data are provided with this paper.

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

## Acknowledgements

This work was funded by European Union's Horizon 2020 research and innovation programme under GA 952334 (PhasAGE), by the Spanish Ministry of Science and Innovation (PID2019–105017RB-I00) to S.V. and by ICREA, ICREA-Academia 2015 and 2020 to S.V. J.G-P. was supported by the Spanish Ministry of Science and Innovation with a Juan de la Cierva Incorporación IJC2019-041039-I. M.G-G. was supported by the Spanish Ministry of Science and Innovation via doctoral grants, FPU16/02465. S.R. was funded by Fondazione ARISLA (project TDP- 43-STRUCT) and by Ricerca Corrente funding from Italian Ministry of Health to IRCCS Policlinico San Donato. The authors thank the Microscopy Service, Universitat Autònoma de Barcelona, for their exceptional technical support in electron microscopy and confocal imaging, and the University of Milan Unitech NOLIMITS Center for granting access to the cryo-EM facility. We thank Prof. J. Paul Taylor for providing the hnRNPDL KO HeLa cell line and Dr. Tim Schulte for technical assistance.

## Author contributions

J.G-P., M.B., S.R. and S.V. designed the conceptual framework of the study and experiments. J.G-P., A.B-N., M.G-G. and C.V. performed the experiments and contributed to data acquisition, data analysis, and preparation of manuscript figures. A.C-S., J.G.-P., and M.G-G. contributed to cryo-EM data acquisition and cryo-EM structural characterization of the amyloid fibrils. J.G-P. and S.V. wrote the manuscript with contributions and comments from all authors.

## Competing interests

The authors declare no competing interests.
