## [Peer Review File · Nature Communications]

REVIEWER COMMENTS

Reviewer #1 (Remarks to the Author):

The authors determined the cryoEM structure of hnRNPD-2, a ribonucleoprotein involved in transcription and RNA-processing, in which missense mutations in a single residue in the exon 6 domain are associated with muscular dystrophy (LGMDD3). The ordered part of the fibril, for which atomic resolution structure was obtained, contains residues 226-276, encoding for a region named exon 6, indicating its role in fibril formation. Two consecutive globular RNA recognition motifs (RRM1 and RRM2), located at the N-terminal part, which bind DNA/RNA, are partially disordered, and could not be determined at high resolution, yet 2D classes suggest that they decorate the fibril core. The C-terminal part (21 residues), comprised of a nuclear localization sequence, is disordered.

The manuscript is well written, relates to previous findings in the field, and provides comprehensive support for the interpretation of the structure with functional implications. The findings overall contribute to the array of functional amyloids, and to potential differences compared to toxic and pathological amyloids. The lack of polymorphism in the fibril core is interesting, but the disordered regions do suggest polymorphism/disorder in the decoration of the fibril core. The manuscript also implies potential evolutionary functional considerations for the utilization of phase separation vs fibril formation. In addition, the manuscript suggests that a disease-causing mutation might be related to loss-of-function by preventing fibrillation, which is rather exceptional.

Comments:

1. The abstract is somewhat misleading in terms of what was practically observed. I mean the sentence: "We present the cryo-electron microscopy structure of full-length hnRNPD-2 amyloid fibrils, which are stable, non-toxic, and bind nucleic acids, with the RNA binding domains building a solenoidal coat around them". I would be clearer about the modelled atomic resolution part of the exon 6 domain vs predictions from the 2D classes related to the decoration.
2. Fig 1 panel b – please explain the image including green/blue colours and marks. The same for Extended Data Fig. 3.
3. Extended Data Fig. 2 – please explain in the figure legend or text what is observed. The text mention transfection but the legend does not explain it and what exactly are we looking at.
4. Table 1 indicates 255 protein residues and 50 water molecules. Is that for 4 layers?
5. The sentence: "48 % of which is covered by polar atoms, a value that is significantly lower than in disease-associated fibrils" – did you mean "significantly higher"?
6. Supplementary Tables 2/3 – SASA calculations – What about the % of polar residues in the buried area? Functional amyloids were assumed to have a more liable cross-beta interfaces with reduced stability compared to pathological amyloids, which manifest with higher % of glycine and polar residues.
7. Supplementary Tables 2/3 – SASA calculations – It is important to take into account that the resolved parts of the structures of Abeta and IAPP represent the full-length proteins, while the other structures are of fragments / ordered parts. The rest of the protein, even though not ordered, might completely change the nature of the surface of the fibril.
8. You still have a comment unresolved in the SI file (Extended Data Fig. 12.).
9. Fig 3e – G258 should be Y258.
10. Fig 3c – the electrostatics seems very rough (full charges? I could not find the description in the method section). Please indicate the charge scale used. If possible, I would calculate with partial charges with a reliable force field to get a better view of the negative surface. The region between the RRM1 and the exon 6 seems positively charged and can compensate the negative interface. However, if electrostatic interactions do occur, one might expect a more ordered conformation of this region to be seen in the map. Perhaps a structure in different salt concentrations might allow additional views on a larger

fragment (This reviewer is not expecting this in the revisions – just to be clear..).

11. Only thoughts, no comments: Supplementary Table 4 and Extended Data Fig. 11 - differences in the solvation free energy of the buried cores – this calculations is very much dependent on the SASA as far as I remember. Thus, differences in the polar/electrostatics which effect enthalpy are somewhat neglected. Still, with the lack of a hydrophobic fibril core, where is the thermostability coming from, and can electrostatics hold? Maybe the n-n stacking. The role of the additional unmodeled domains is still unknown and might provide a reasonable energetic explanation. Anyway, “skipping” phase separation compared to hnRNPA1/A2 LCDs fits the thermostability of the hnRNPD-2 fibrils.

Extended Data Fig. 15 – I would add a graph showing how you determined the Kd. How was the RNA sequence chosen? How was the ssDNA fragment length and sequence for the binding assay chosen? Can it be that the affinity is sequence specific and not about DNA/RNA comparisons?

12. Fig. 4h “Binding of the 7-mer fluorescent ssDNA (F-ssDNA) to hnRNPD-2 amyloid filaments”. The X-axis shows something different. Can you also provide the gel too?

13. While it is more intuitive to understand how phase separation and membraneless organelles can protect RNA, here, it is hard to understand the role of fibril formation in the regulation/protection of RNA/DNA. The sentence in the Discussion: “The most substantial evidence of hnRNPD-2 fibrils' functionality is their ability to bind oligonucleotides, especially ssDNA, with an affinity equal to or higher than the soluble counterpart.” Can affinity to soluble vs fibril really be compared considering fast fibrillation? The DNA/RNA can also affect fibrillation rate and fibril morphology/stability. Were the thermostability measurements done in the presence of any DNA/RNA? (even coming from the expressing cell)

Reviewer #2 (Remarks to the Author):

The work presented by Garcia-Pardo et al., presents exciting and novel discoveries of interest to the RNP-amyloid field, and, as this is a functional amyloid form, with broader interest to the field of protein folding. They describe the first hnRNPD amyloid fibril structure, which is said to be the first structure of a full-length RNP in its functional conformation. This claim appears to be justified and strongly supported from the structural analysis within the paper: the fibril core matches to exon 6 - which alternative splicing removes from the mammalian isoform hnRNPD-3. Interestingly, the nuclear-localisation sequence (PY-NLS) is not buried within the fibril core as was suggested by previous studies of a truncated form of hnRNPA1/A2 LCDs. Garcia-Pardo et al also show that the RNA-binding RRM domains are present on the surface of the fibril and that the fibrils are able to bind nuclei acid (specifically short ssDNA fragments and, more weakly, RNA. In addition, the authors use their new structure to rationalise disease-associated mutations, supported by biochemical evidence that these mutants do not form fibrils in vitro under the same conditions used to fibrillate the WT protein. Overall, the experiments and interpretations present a cohesive, relatively complete and informative story that is suitable for publication in Nature Communications, subject to the suggested revisions, as discussed below.

Technically the work described is well carried out, but there are some aspects that raised questions. Firstly, the fact that hnRNPD-1, hnRNPD-3, hnRNPD-2(D259H) and hnRNPD-2(D259N) do not form amyloid fibrils is a key part of many of the discussion points within the manuscript. Given the large variation in fibril formation ability observed by the WT hnRNPD-2 protein when the concentration of NaCl is varied in the reaction setup (ED Fig.5), different conditions or lengths of time may be required for the four variants to assemble into amyloid. The authors should test different salt concentrations for growth of

the variant fibrils, and assay over much longer times e.g. up to 1 week. ED Fig.4b only shows a ThT reaction up to 24 hours and there seems to be a significant increase in ThT fluorescence for hnRNPD1-1. The data reported are the average of three repeats. The authors must show each individual trace so that reproducibility can be seen clearly, if possible using more replicates. Additionally, for the two point mutants, no ThT experiments are reported to support the lack of fibrillar species seen by sedimentation. This should be included.

I was also puzzled by the author's choice of short ssDNA for testing binding to the functional fibrils of hnRNPD1, and the observation that the affinity for RNA for this supposed RNA-binding fibril is so weak. Why were such short fragments used, and would an increase in affinity be observed for longer RNA sequences because of avidity?

Other minor points and revisions:

- The α -Syn and A β structures selected to represent pathogenic fibrils are both in vitro generated structures that differ from solved ex vivo structures. Given that the authors are comparing functional versus pathogenic amyloids, they should report the comparisons using the known, pathogenic fibril structures for these proteins. It would be helpful to show more example structures in these comparative analyses, as the results section makes claims of hnRNPD1-2 fibrils against all pathogenic fibrils.
- Lines 211-213, need correcting to read: "a value that is significantly lower in disease-associated fibrils..." instead of "significantly lower than in...".
- Extended Data Fig. 9d, there's a water H-bond that doesn't appear to link to another visible atom?
- Missing citation for Coot in the Materials and Methods.
- Please check over the figure legends, in several cases there is missing information e.g. what are the arrows pointing to in Fig. 1b, what is blue and what is green? What are the red dots in Fig. 2 (this only becomes clear once ED figures are looked at)? It would be helpful for consistent colours also to be used in Fig. 1a and ED Fig. 1.
- Could Fig. 1a be improved? Which residues correspond to exon 6 and how is the numbering reported for the different constructs with different N-terminal lengths.

Reviewer #3 (Remarks to the Author):

Overview of relevant issues of the research:

This study brilliantly describes the full length fibrillar structure of hnRNPD1 by cryo-electron-microscopy. This member of the ribonucleoprotein (RNP) family shares a prion domain conferring aggregation attributes and nucleic acid binding domains relevant for biogenesis and metabolism of RNA.

Pathological mutations in several RNPs give rise to neurodegenerative diseases involving CNS, motor neuron, peripheral nerve and muscle that may appear as an isolated disorder (e.g. ALS) or as a multisystemic impairment.

Regarding the hnRNPD1 disease-causing mutations only myopathic affectation (LGMD-D3) has been reported. Pathologically, the affected cells display insoluble deposits and dislocation of the altered riboprotein usually accompanied by other related protein partners, along with signs of activation of diverse metabolic cell routes acting in defense or as expression of the degenerative cascade. Currently there are many gaps in the understanding of that complex scenario.

The study reveals that the hnRNPD1-2 isoform in its fibrillar conformation holds a particular

structure different from other closely related RNPs (hnRNPA1, hnRNPA2), permitting to maintain the functional capabilities of nucleic acid processing while their prion amyloidogenic propensity and cytotoxic abilities are limited. An extensive physical entropic analysis of the peptide model gives a rational explanation for the behavior of this construct. The authors propose that a physiologic equilibrium between the hnRNPD2-2 monomeric and fibrillar conformation may be relevant for their connection with RNA metabolism, topologic distribution, and putative interaction with other proteins.

Importantly, analyses of the disease-related mutations yielded abnormal conformed peptide variants that similarly to the wt. lacked fibrillar aggregation propensity. This led the authors to postulate the existence of alternative patho-mechanisms to the prevailing thesis of insoluble phase transition / amyloidogenic deposition, which might include loss of function by a quick elimination of the mutated proteins via proteosome / autophagic routes or deleterious effects of the mutated species.

The authors appeal to the absence of protein aggregates in muscles from LGMD-3 patient´s biopsies or mutated fly model supporting this hypothesis.

Other interesting contribution of this research deal with the description of the molecular structure and functional behavior of the three hnRNPD2 isoforms using a transfected cell model and in vitro protein analysis.

Comments and concerns:

This work significantly contributes to the understanding of the complex matter of ribonucleoprotein pathophysiology. The paper is well written, the experimental procedures clearly described, and figures and table of data are highly illustrative.

However, there are some points worth mentioning:

1- Previous experiments indicates that hnRNPD2mutants exert a cell toxic effect (Navarro et al 2015). How can you interpret the discrepancy with the current results?

2- Certainly, several LGMD-3 myopathologic studies coincide in the absence of sarcoplasmic protein aggregates (Viera et al 2014, Sun 2019, Berardo 2019). However, amyloid fibril / congophilic deposits were detected in some instances (Viera et al 2014, Vicente et al 2020). A remark on this issue would be of interest.

3- In Viera´s study TPNO-1 co-localized with hnRNPD2 in nucleus in normal control but the pattern changed in mutated giving a variable nuclear / perinuclear pattern. Though the issue is not addressed in this work, would it be of interest to make a quick remark on the potential implications of that interaction?

4- Are there any hypothetical explanation why deleterious hnRNPD2 mutations are restricted to muscle ?

5- I miss proposal delving into the alternative or concurrent patho-mechanisms implicated in LGMD-D3 to be undertaken on patient´s muscle / cultivate cells, and disease models.

Reviewer 1

REVIEWER COMMENTS

Reviewer #1 (Remarks to the Author):

The authors determined the cryoEM structure of hnRNPD-2, a ribonucleoprotein involved in transcription and RNA-processing, in which missense mutations in a single residue in the exon 6 domain are associated with muscular dystrophy (LGMDD3). The ordered part of the fibril, for which atomic resolution structure was obtained, contains residues 226-276, encoding for a region named exon 6, indicating its role in fibril formation. Two consecutive globular RNA recognition motifs (RRM1 and RRM2), located at the N-terminal part, which bind DNA/RNA, are partially disordered, and could not be determined at high resolution, yet 2D classes suggest that they decorate the fibril core. The C-terminal part (21 residues), comprised of a nuclear localization sequence, is disordered.

The manuscript is well written, relates to previous findings in the field, and provides comprehensive support for the interpretation of the structure with functional implications. The findings overall contribute to the array of functional amyloids, and to potential differences compared to toxic and pathological amyloids. The lack of polymorphism in the fibril core is interesting, but the disordered regions do suggest polymorphism/disorder in the decoration of the fibril core. The manuscript also implies potential evolutionary functional considerations for the utilization of phase separation vs fibril formation. In addition, the manuscript suggests that a disease-causing mutation might be related to loss-of-function by preventing fibrillation, which is rather exceptional.

Answer from authors: We thank Reviewer 1 for the very positive and encouraging comments. We tried to address and clarify all the commentaries in the following lines.

Comments:

1. The abstract is somewhat misleading in terms of what was practically observed. I mean the sentence: "We present the cryo-electron microscopy structure of full-length hnRNPD-2 amyloid fibrils, which are stable, non-toxic, and bind nucleic acids, with the RNA binding domains building a solenoidal coat around them". I would be clearer about the modelled atomic resolution part of the exon 6 domain vs predictions from the 2D classes related to the decoration.

Answer from authors: The reviewer is correct. Although we have solved the structure corresponding to exon 6 at high resolution (2.5 Å), the RNA binding domains were located based on the information from the 2D class averages. Experimental attempts to solve the structure of the globular domains at high resolution were made, but they proved impossible due to lack of well defined positions of the domains relative to the

fibril structure, translating in fuzzy-very weak density. Therefore, our ability to understand their structure and interactions with the amyloid core has been limited to the predictive models. To clarify this point, as suggested, we have changed the abstract in the revised version of the manuscript (see changes on page 2).

2. Fig 1 panel b – please explain the image including green/blue colours and marks. The same for Extended Data Fig. 3.

Answer from authors: We thank the reviewer for the suggestion. Fig. 1b and Extended Data Fig. 3 show representative confocal images of HeLa hnRNPD L KO cells transiently transfected with each of the different GFP-tagged hnRNPD L isoforms (DL-1, DL-2, and DL-3) or with the empty GFP vector (GFP). The images show the characteristic nuclear distribution of these proteins in green and the nuclear DNA, stained with Hoechst, in blue. Moreover, Extended Data Fig. 3 also shows the regions of interest (ROI) used for the fluorescence intensity analyses. These regions are depicted as yellow dashed lines over the confocal images. The nucleus contour has been highlighted in all cases with a white dashed line. According to the reviewer's suggestion, we have now included this information in the new versions of these figures, which have been included in the figure legends of Fig. 1b (page 6) and Extended Data Fig. 3 (see revised Extended Data file, page 8).

3. Extended Data Fig. 2 – please explain in the figure legend or text what is observed. The text mention transfection but the legend does not explain it and what exactly are we looking at.

Answer from authors: We appreciate the reviewer's suggestion. We are sorry if we did not make it clear that Extended Data Fig. 2 shows the characterization of the HeLa hnRNPD L knockout (KO) cell line used in the present study. The left panel corresponds to a western blot showing the levels of endogenous hnRNPD L in wild-type HeLa cells (HeLa WT) compared with hnRNPD L knockout cells (hnRNPD L KO). The presence of the protein in the soluble fraction was detected using a specific anti-hnRNPD L commercial antibody (see Material and Methods section for experimental details). In the right panel, we have included a Ponceau S staining of the same blot as a loading control. As suggested by the reviewer, we have included this previously missed information in the revised version of the Extended Data Fig. 2 legend (see revised Extended Data file, page 7).

4. Table 1 indicates 255 protein residues and 50 water molecules. Is that for 4 layers?

Answer from authors: We thank the Reviewer for this question. Supplementary Table 1 summarized the cryo-EM data collection and refinement statistics for the hnRNPD amyloid structure. We are sorry if we did not make it clear that this structure includes 5 layers, which correspond to five individual chains. In total, they account for 255 protein residues and 50 water molecules. This is a relevant parameter, and we have now included this information in the revised version of Supplementary Table 1 as an additional note (see revised Extended Data file, page 3).

5. The sentence: “48 % of which is covered by polar atoms, a value that is significantly lower than in disease-associated fibrils” – did you mean “significantly higher”?

Answer from authors: We thank the reviewer for pointing out the error. The solvent-accessible surface area (SASA) of the hnRNPD-2 fibril upper layer is covered by 48% of polar atoms, a value significantly higher than those observed for disease-associated fibrils whose values range from 28 to 32%. Accordingly, we have replaced “significantly lower” with “significantly higher” in the revised manuscript, page 10.

6. Supplementary Tables 2/3 – SASA calculations – What about the % of polar residues in the buried area? Functional amyloids were assumed to have a more liable cross-beta interfaces with reduced stability compared to pathological amyloids, which manifest with higher % of glycine and polar residues.

Answer from authors: We thank the reviewer for the question. This is an excellent point, and we have performed new calculations to determine the percentage of polar residues in the buried area of the amyloids' inner and upper layers; in Supplementary Tables 2 and 3. Interestingly, we found that, on average, functional amyloids manifest a significantly higher percentage of polar buried residues than pathological amyloids. Similar behaviour was observed for the external and inner layers of the amyloid fibril. It is worth mentioning that the hnRNPD-2 amyloid fibril showed the highest percentage of polar buried residues (64% and 67% for the upper and inner layers, respectively) of all the amyloid structures analysed. This new data and additional discussion regarding this point have been included in the revised Extended Data file, page 4 (see new Tables 2 and 3 below) and in the revised manuscript, pages 10 and 11.

Revised Extended Data Table 2 and 3:

Supplementary Table 2. Solvent-accessible surface area (SASA) parameters calculated for the upper layers of hnRNPD-2, hnRNPA2, hnRNPA1, hSAA1, A β -42 and α -synuclein amyloid fibrils.

	Total Upper Surface (Å ²)	Buried Upper Surface (Å ²)	Exposed Upper Surface (Å ²)	Exposed Polar Surface (%)	Polar Buried Residues (%)
hnRNPD-2	5789	2330	3458	48	64
hnRNPA1	4721	2480	2241	42	47
hnRNPA2	5761	2382	3379	42	52
hSAA1	6482	3078	3404	38	47
α -synuclein	6168	2521	3648	36	42
A β -42	3644	1706	1938	32	31

Note that proteins are ordered according to decreasing Exposed Polar Surface (%) values. Exposed Polar (%) values were calculated using GetArea¹. Surfaces values and polar buried residues (%) were calculated with PDBePISA². PDB accession codes used for calculations are: hnRNPD-2 (PDB 7ZIR, residues 345-395 according to hnRNPD-1 numbering), hnRNPA2 (PDB 6WQK, residues 263-319), hnRNPA1 (PDB 7BX7, residues 251-295), hSAA1 (PDB 6MST, residues 2-55), A β -42 (PDB 5KK3, residues 11-42) and α -synuclein (PDB 6OSJ, residues 37-97).

Supplementary Table 3. Solvent-accessible surface area (SASA) parameters calculated for the inner layers of hnRNPD-2, hnRNPA2, hnRNPA1, hSAA1, A β -42 and α -synuclein amyloid fibrils.

	Total Inner Surface (Å ²)	Buried Inner Surface (Å ²)	Exposed Inner Surface (Å ²)	Exposed Polar Surface (%)	Polar Buried Residues (%)
hnRNPD-2	5761	4467	1294	57	67
hnRNPA2	5758	4518	1240	47	53
hnRNPA1	4721	4117	604	44	46
hSAA1	6477	5474	1003	43	46
A β -42	3589	2868	720	40	31
α -synuclein	6170	4741	1429	37	42

Note that proteins are ordered according to decreasing Exposed Polar Surface (%) values. Exposed Polar (%) values were calculated using GetArea¹. Surfaces values and polar buried residues (%) were calculated with PDBePISA². PDB accession codes used for calculations are: hnRNPD-2 (PDB 7ZIR, residues 345-395 according to hnRNPD-1 numbering), hnRNPA2 (PDB 6WQK, residues 263-319), hnRNPA1 (PDB 7BX7, residues 251-295), hSAA1 (PDB 6MST, residues 2-55), A β -42 (PDB 5KK3, residues 11-42) and α -synuclein (PDB 6OSJ, residues 37-97).

7. Supplementary Tables 2/3 – SASA calculations – It is important to take into account that the resolved parts of the structures of Abeta and IAPP represent the full-length proteins, while the other structures are of fragments / ordered parts. The rest of the protein, even though not ordered, might completely change the nature of the surface of the fibril.

Answer from authors: The reviewer is completely right. We agree that there are differences in the fraction of the sequence covered by the structures of A β -42 (PDB 5KK3) and hSAA1 (PDB 6MST) and those of α -synuclein (PDB 6OSJ) and the hnRNPs hnRNPA1 (PDB 7BX7), hnRNPA2 (PDB 6WQK) and hnRNPD-2 (PDB 7ZIR). We agree that such structurally unsolved extensions may affect the amyloidogenic properties of the protein, as well as tune the nature of the surface of the amyloid fibril. Unfortunately, considering these regions for our calculations is currently not feasible due to the lack of structural information. Alternatively, we have specified in the revised Tables which are the residues from each protein considered for the calculations (Supplementary Tables 2 and 3 in the revised Extended Data file on page 4).

8. You still have a comment unresolved in the SI file (Extended Data Fig. 12.).

Answer from authors: We thank the reviewer for pointing out the unresolved comment. We have removed this from the revised SI file.

9. Fig 3e – G258 should be Y258.

Answer from authors: We thank the reviewer for pointing out the error in Fig 3e. We have replaced “G258” with “Y258” in the revised version of Fig. 3e (page 12 of the revised manuscript).

10. Fig 3c – the electrostatics seems very rough (full charges? I could not find the description in the method section). Please indicate the charge scale used. If possible, I would calculate with partial charges with a reliable force field to get a better view of the negative surface. The region between the RRM and the exon 6 seems positively charged and can compensate the negative interface. However, if electrostatic interactions do occur, one might expect a more ordered conformation of this region to be seen in the map. Perhaps a structure in different salt concentrations might allow additional views on a larger fragment (This reviewer is not expecting this in the revisions – just to be clear..).

Answer from authors: Surface electrostatic potential maps were re-calculated using the PDB2PQR/APBS web servers and visualized using PyMol (Schrödinger, NY, USA)³. Protonation states were assigned at pH 7, and the corresponding electrostatic potential maps are plotted in $k_bT e_c^{-1}$, with the Boltzmann's constant k_b , the charge of an electron e_c , and a temperature T of 298 K. Electrostatic potential maps using Amber or Parse force fields were visually indistinguishable from each other.

As shown in the new version of Fig. 3c, the surface of the amyloid exon 6 is highly negative. As suggested by the reviewer, this region may establish interactions with the positively charged linker between the RRM domains and the amyloid core. In support of this hypothesis, we observed additional low-resolution densities in the unsharpened cryo-EM map (see Fig. 4a-d), which could fit with the partial ordering of this linker region and/or with the RRM domains. We also observed a significant dependence of its aggregation on the salt concentration (Extended Data Fig. 5), suggesting that such electrostatic contacts may play an important role in modulating this interaction. However, despite this experimental evidence, solving the structure at different salt concentrations would be extremely challenging. At present we are using SAXS and molecular dynamics to try to understand the residues responsible for this dependence, but this would be the subject of a subsequent work.

Fig. 3c

Fig. 3. Overall structure of the hnRNPD2-2 fibril core. c) Surface representation showing the electrostatic surface potential distribution of the hnRNPD2-2 fibril with ribbon representation of one subunit on top. Partial charges and electrostatic potential maps were calculated using APBS server and visualized using the APBS plugin in PyMol (Schrödinger, NY, USA)^{3,4}. Protonation states were assigned at pH 7. Negative and positive charge potentials are colored in red and blue, respectively. The location of the six β -strands ($\beta 1$ to $\beta 6$) is indicated. The side chain of D259, mutated in LGMDD3, is shown as blue sticks, and labeled.

11. Only thoughts, no comments: Supplementary Table 4 and Extended Data Fig. 11 - differences in the solvation free energy of the buried cores – this calculations is very much dependent on the SASA as far as I remember. Thus, differences in the polar/electrostatics which effect enthalpy are somewhat neglected. Still, with the lack of a hydrophobic fibril core, where is the thermostability coming from, and can electrostatics hold? Maybe the π - π stacking. The role of the additional unmodeled domains is still unknown and might provide a reasonable energetic explanation. Anyway, “skipping” phase separation compared to hnRNPA1/A2 LCDs fits the thermostability of the hnRNPD-2 fibrils.

Answer from authors: The reviewer is correct that the previous ΔG^{int} values, indicating the solvation free energy gain for the respective structure upon assembly formation, were calculated as the difference in total solvation energies of the isolated structure and the same structure as part of the assembly. As correctly stated by the reviewer, this value does not include the effect of satisfied hydrogen bonds and salt bridges across the interfaces. Now, we have calculated ΔG , which indicates the solvation free energy of folding for the corresponding structure, in kcal/mol and include the contribution of those interactions, obtaining, as expected, higher stability values for all the analyzed structures. At the request of reviewer #2, we have now included, in Supplementary Table 4 and Extended Data Fig. 13, calculations for other pathogenic fibrils, including some obtained *ex vivo*, resulting in the analysis of a total of 17 fibril structures. As stated previously, the data indicate that hnRNPs fibrils are clearly less stable than pathogenic ones. As the reviewer affirms, π - π stacking might also contribute significantly to the fibril stability, as well as contacts with adjacent linkers or RRMs, although these contributions cannot be calculated appropriately. The results and discussion sections have been slightly modified to incorporate these new data (pages 11, 18).

Extended Data Fig. 15 – I would add a graph showing how you determined the K_d .

Answer from authors: We thank the reviewer for the suggestion. We have included an additional graph in Extended Data Fig. 18 (new numbering in the revised version of the Extended Data file) with the densitometric analysis of the electrophoretic mobility shift assay (EMSA) of the RNA binding to hnRNPD-2 (see the revised version of Extended data Fig. 18 below). We have also provided the K_d value determined for the interaction of hnRNPD-2 with RNA, which is $22.8 \pm 3.95 \mu\text{M}$. The new data is presented below and included in the Manuscript and Extended Data File revised versions (pages 17 and 23, respectively).

Extended Data Fig. 18

Extended Data Fig. 18. RNA (F-GACUAGC) binding to soluble hnRNPDL-2. a) Electrophoretic mobility shift assay (EMSA) of soluble hnRNPDL-2 with a Fluorescein-labelled RNA (F-GACUAGC). The 7-mer RNA was incubated with increasing concentrations of the soluble form of hnRNPDL-2. The assayed protein concentrations are indicated in μM. b) Binding affinity of soluble hnRNPDL-2 to the 7-mer fluorescent RNA (F-RNA) determined by the EMSA assay. Band intensities from three independent EMSA assays were quantified and used to calculate the K_d value shown in b) by assuming a one-site specific binding mechanism with Hill slope (see Materials and Methods section for equation details). In b), data is shown as mean ± SEM (n = 3 independent experiments).

How was the RNA sequence chosen? How was the ssDNA fragment length and sequence for the binding assay chosen? Can it be that the affinity is sequence specific and not about DNA/RNA comparisons?

Answer from authors: These are very pertinent questions. DNA/RNA binding is thought to be mediated by the two RNA recognition motifs (RRMs) in this family of proteins. The RRM fold is a prevalent RNA-binding domain that consists of approximately 80-90 amino acids and binds to single-stranded RNA/DNA. The RRM of different proteins have a variety of RNA binding preferences and functions. Accordingly, we agree with the reviewer that the affinity of hnRNPDL-2 is likely sequence specific.

We could only find one study in which the specificity of hnRNPDL-2 RRM has been studied⁵, and we selected the RNA sequence based on these findings. We are sorry that we did not clarify this information in the first version of the manuscript. We have now clarified in the revised manuscript (page 17) that we have chosen our RNA/DNA sequence based on a previous publication in which the authors studied the interactions between JKTBP (an alternative name for hnRNPDL) and different RNA sequences. In this study, the authors proposed that the consensus binding site was ACUAGC; such

consensus sequence was derived from a screening with RNA ligands selected from a pool of 20 nt random sequences inserted into 74-mer oligoRNAs by selection/amplification. Thus, we decided to first evaluate the affinity of the hnRNPD-2 to RNA oligonucleotides containing the ACUAGC motif. However, we found that the binding of a FITC labelled 7-mer RNA with this consensus sequence showed a relatively low affinity with a K_d value of about $22.8 \pm 3.95 \mu\text{M}$ (see Extended Data Fig. 18). For this reason, we also evaluated the binding of the protein to an equivalent sequence of DNA, as shown in the revised Fig. 4f and Fig. 4g, which showed a K_d of about $5.9 \pm 0.78 \mu\text{M}$. One possibility is that in addition to the ACUAGC consensus sequence, different length and/or additional nucleotides are needed to achieve a higher binding affinity. Unfortunately, specific long RNA/DNA hnRNPD targets remain to be identified. This reflection has been introduced in page 17.

12. Fig. 4h “Binding of the 7-mer fluorescent ssDNA (F-ssDNA) to hnRNPD-2 amyloid filaments”. The X-axis shows something different. Can you also provide the gel too?

Answer from authors: The reviewer is right in this observation. We would like to mention that the hnRNPD-2 fibril filaments cannot be resolved in an EMSA gel. For this reason and to evaluate their binding to DNA, we measured the fluorescence of the fibrils after incubating them with different concentrations of the fluorescein-labelled ssDNA. This assay allowed us to determine an approximate K_d for the binding of about $2.1 \pm 0.36 \mu\text{M}$. Finally, this interaction was further confirmed by directly visualizing the fluorescence of the fibers in a fluorescent confocal microscope (see Fig. 4i).

We are sorry if we did not make clear in the first version of the manuscript that we followed a different experimental approach to evaluate the binding of the 7-mer fluorescent ssDNA to hnRNPD-2 amyloid fibrils. This has been clarified in the revised manuscript (page 17). In addition, we have also provided more accurate K_d calculations after including additional replicates and analyzing the data following a one-site specific binding mechanism (See revised Fig. 4h). The new K_d value is reported in the revised manuscript on page 17. Finally, following a suggestion from the second reviewer, we have included additional replicates (at least $n=3$) for all the RNA/DNA binding experiments. These figures and the derived K_d values in the text have been updated accordingly. Additionally, in the discussion we have changed the sentence “...with an affinity equal or higher than that of the soluble counterpart.” to “...with an affinity equivalent to that of the soluble counterpart.”, which we agree is more accurate (page 21).

13. While it is more intuitive to understand how phase separation and membraneless organelles can protect RNA, here, it is hard to understand the role of fibril formation in the regulation/protection of RNA/DNA. The sentence in the Discussion: “The most

substantial evidence of hnRNPD-2 fibrils' functionality is their ability to bind oligonucleotides, especially ssDNA, with an affinity equal to or higher than the soluble counterpart." Can affinity to soluble vs fibril really be compared considering fast fibrillation? The DNA/RNA can also affect fibrillation rate and fibril morphology/stability. Were the thermostability measurements done in the presence of any DNA/RNA? (even coming from the expressing cell)

Answer from authors: This is an excellent point. We understand the reviewer's concern, and we agree that the affinity for soluble vs. fibrils states cannot be directly compared, and this has been made clear in the revised manuscript. We would like to clarify that our initial aim was to demonstrate that the RRM domains retained a folded conformation in the fibrillar state of the protein since the common assumption is that they misfold during aggregation and that the protein totally loses any function. Surprisingly, this assumption has not been sustained by any experimental evidence. This is why we tested the ability of the fibrils to bind nucleic acids, which we firmly believe is a clear indication of their functionality.

It has been previously suggested that AS might regulate the assembly properties of RNA-processing proteins by controlling the incorporation of multivalent disordered regions in the isoforms⁶, but this has been assumed to occur through phase separation phenomena. In this study, we have demonstrated that an amyloid-specific alternative exon is able to regulate the assembly of hnRNPD-2 by modulating fibril formation. Furthermore, we have also proven the ability of such ordered assemblies to bind oligonucleotides, although we can only hypothesize that they will be functional in the cellular context.

As suggested by the reviewer, an interesting possibility is that RNA might modulate the assembly of hnRNPD-2 into amyloids. To address this question, we have performed new aggregation reactions of hnRNPD-2 in the absence and presence of RNA. This has not been trivial because Thioflavin-T becomes highly fluorescent in the presence of oligonucleotides, which impedes any kinetic measurement. After unsuccessfully trying different dyes for this experiment, we turned to commercial Proteostat[®] to monitor the aggregation of hnRNPD-2 in the absence and presence of RNA (see figure below). As it can be seen, when using a 1.8:1 protein/RNA ratio, the nucleic acid impacts the final Proteostat[®] fluorescence and the slope of the kinetic curve, but this effect is moderate, suggesting that exon 6 can drive the assembly independently of the presence of RNA.

Our idea is to try solving the structure of the resulting fibrils by cryo-EM and the ones generated by adding RNA to preformed fibrils, but this will be the subject of a subsequent study; in which we will also include the experiment suggested by the reviewer, since we think it fits better in that context than in the present work.

Figure for reviewers. Effect of RNA on hnRNPDL-2 aggregation. Aggregation kinetics of soluble hnRNPDL-2 in the absence (red dots) and presence of 10 μ M Baker's yeast tRNA (black dots). 10 μ M Baker's yeast tRNA was used as a control (grey dots). Aggregation is shown as Proteostat[®] binding over time. Samples were incubated at 37 $^{\circ}$ C and under constant shaking in 96 well plates as described for other aggregation experiments. Values are shown as mean \pm SEM, n =2 independent experiments. Left and right panels show absolute and normalized fluorescence values, respectively.

Reviewer 2

REVIEWER COMMENTS

Reviewer #2 (Remarks to the Author):

The work presented by Garcia-Pardo et al., presents exciting and novel discoveries of interest to the RNP-amyloid field, and, as this is a functional amyloid form, with broader interest to the field of protein folding. They describe the first hnRNPD L amyloid fibril structure, which is said to be the first structure of a full-length RNP in its functional conformation. This claim appears to be justified and strongly supported from the structural analysis within the paper: the fibril core matches to exon 6 - which alternative splicing removes from the mammalian isoform hnRNPD L-3. Interestingly, the nuclear-localisation sequence (PY-NLS) is not buried within the fibril core as was suggested by previous studies of a truncated form of hnRNAPA1/A2 LCDs. Garcia-Pardo et al also show that the RNA-binding RRM domains are present on the surface of the fibril and that the fibrils are able to bind nuclei acid (specifically short ssDNA fragments and, more weakly, RNA. In addition, the authors use their new structure to rationalise disease-associated mutations, supported by biochemical evidence that these mutants do not form fibrils in vitro under the same conditions used to fibrillate the WT protein. Overall, the experiments and interpretations present a cohesive, relatively complete and informative story that is suitable for publication in Nature Communications, subject to the suggested revisions, as discussed below.

Answer to the reviewer: We sincerely thank the reviewer for the constructive criticism and helpful suggestions. We hope that the responses below and the edits made to the text respond adequately his/her comments.

Technically the work described is well carried out, but there are some aspects that raised questions. Firstly, the fact that hnRNPD L-1, hnRNPD L-3, hnRNPD L-2(D259H) and hnRNPD L-2(D259N) do not form amyloid fibrils is a key part of many of the discussion points within the manuscript. Given the large variation in fibril formation ability observed by the WT hnRNPD L-2 protein when the concentration of NaCl is varied in the reaction setup (ED Fig.5), different conditions or lengths of time may be required for the four variants to assemble into amyloid. The authors should test different salt concentrations for growth of the variant fibrils, and assay over much longer times e.g. up to 1 week. ED Fig.4b only shows a ThT reaction up to 24 hours and there seems to be a significant increase in ThT florescence for hnRNPD L-1.

Answer from authors: We agree with the reviewer that the observation that hnRNPD L-1 and hnRNPD L-3 do not form amyloid fibrils is a relevant part of our manuscript. According to the reviewer's suggestion, we have performed additional experiments to provide additional support to the previous data. First, we performed new aggregation

kinetics with purified hnRNPD-1, hnRNPD-2 and hnRNPD-3 at different salt concentrations (*i.e.*, 50, 300, and 600 mM NaCl). These experiments confirmed that only the hnRNPD-2 could form amyloid fibrils after 24 hours of incubation. As expected, the highest amyloid fibril formation for hnRNPD-2 was observed in the presence of 300 mM NaCl (see new Figure below, incorporated as Extended Data Fig. 5 on page 10).

Extended Data Fig. 5. Effect of NaCl on hnRNPD-1, hnRNPD-2 and hnRNPD-3 aggregation. a-c) Aggregation kinetics of soluble a) hnRNPD-1, b) hnRNPD-2 and c) hnRNPD-3 at the indicated NaCl concentrations. Aggregation is shown as Th-T binding over time. Samples were incubated at 37 °C and under constant shaking in 96 well plates as described. In a-c), data is shown as mean \pm SEM ($n = 3$ independent aggregation experiments).

We have added these new results and an additional discussion regarding this point in the revised manuscript. Accordingly, Extended Data Fig. 5 has been updated (see changes on Page 10).

According to the reviewer's suggestion, we have also evaluated the aggregation of these three protein variants at longer incubation times. As shown below, incubating the soluble proteins for up to 7 days resulted in similar results, with only hnRNPD-2 showing strong amyloid conversion. We did not detect an increase in the Th-T binding for hnRNPD-1 or hnRNPD-3 after 7 days of incubation. This finding agrees with our previous data and confirms that hnRNPD-1 and hnRNPD-3 do not progress to amyloid aggregates after more prolonged incubation. Finally, this observation was also confirmed by TEM analysis. The results from the new experiments are very interesting since they clearly demonstrate that hnRNPD-1 and hnRNPD-3 isoforms cannot form visible amyloid fibrils even at such extended incubation periods (see new Figure below, incorporated as Extended Data file on page 11).

New experiments for the mutant hnRNPD-2(D259H) and hnRNPD-2(D259N) forms are reported in a subsequent response.

Extended Data Fig. 6. Effect of the incubation time on hnRNPD-1, hnRNPD-2 and hnRNPD-3 aggregation. a-c) Th-T binding to a) hnRNPD-1, b) hnRNPD-2 and c) hnRNPD-3 after incubation of the samples at 37 °C, pH 7.5 and 300 mM NaCl for 0, 2, 5 and 7 days of incubation. Representative TEM micrographs of d) hnRNPD-1, e) hnRNPD-2 and f) hnRNPD-3 after 7 days incubation under the specified conditions.

The data reported are the average of three repeats. The authors must show each individual trace so that reproducibility can be seen clearly, if possible using more replicates.

Answer from authors: According to the reviewer's suggestion, we have now included the errors in all the experiments involving aggregation kinetics to show the reproducibility of each experiment. In some cases, we repeated the experiments to have at least three independent aggregation reactions. We have replaced the corresponding plots and added additional information on Extended Data Fig. 4b (page 9); Extended Data Fig. 5 (page 10). We have also provided relevant statistical information in Extended Data Fig. 8 and 9 legends (page 13 and 14, respectively).

Moreover, we have also included additional replicates (at least $n = 3$) for all the RNA/ssDNA binding experiments. In all the cases, we have specified the type of data shown and the number of replicates performed on the corresponding revised figure legends. These changes have been implemented on the new versions of figures Fig. 4 (page 15) and Extended Data Fig. 18 (page 23).

Additionally, for the two point mutants, no ThT experiments are reported to support the lack of fibrillar species seen by sedimentation. This should be included.

Answer from authors: We appreciate the reviewer's suggestion. We have performed new comparative Th-T binding experiments with the wild type hnRNPD2 and the two-point mutants (D259H and D259N). As shown in the figure below, after setting up aggregation reactions under the same conditions, the two point-mutants showed a weak Th-T fluorescence in comparison with the WT protein. These results agree with our previous data and supports the hypothesis that the two single-point mutants have a reduced aggregation propensity in comparison to the WT protein. This new data (see figure below) has been described (page 14) and incorporated as an additional Extended Data Figure (see Extended Data Fig. 16 in page 21).

Extended Data Fig. 16. Effect of LGMDD3-associated mutations on hnRNPD2 fibril formation. Th-T binding to WT hnRNPD2 (black line) and mutant proteins D259H (red line) and D259N (pink line). All the proteins were incubated at the same conditions (37 °C, pH 7.5 and 300 mM NaCl for 48 h under constant shaking) before the Th-T binding was measured as described in the Materials and Methods section.

I was also puzzled by the author's choice of short ssDNA for testing binding to the functional fibrils of hnRNPD_L, and the observation that the affinity for RNA for this supposed RNA-binding fibril is so weak. Why were such short fragments used, and would an increase in affinity be observed for longer RNA sequences because of avidity?

Answer from authors: We understand the reviewer's concern. As discussed above, in response to the first reviewer, we decided to evaluate the binding of the functional fibrils of hnRNPD_L-2 to a short DNA sequence based on previous data. We could only find one study in which the specificity of hnRNPD_L-2 RRM_s has been studied⁵, and we selected the sequence based on these findings. In this study, the authors proposed that the consensus binding site was ACUAGC; such consensus sequence was derived from a screening with RNA ligands selected from a pool of 20 nt random sequences inserted into 74-mer oligoRNAs by selection/amplification. However, despite choosing this consensus sequence as the target oligonucleotide, the affinity of the RNA was in the low micromolar range.

The data sufficed to confirm that the RRM_s domains retained a folded conformation in the protein's fibrillar state, in contrast to the common assumption that they misfold during aggregation and that the protein totally loses any functionality. However, as suggested by the reviewer, a plausible possibility is that longer sequences could increase the nucleic acid affinity for the fibril. Alternatively, a high-affinity binding might require a combination of specific oligonucleotide sequences binding to the two different RRM_s combined with additional extensions for improving avidity, as shown for other RNA binding proteins. In the absence of a validated long and specific RNA to test, this remains speculative, but these observations are important and have been discussed in the new version of the manuscript (see page 17).

Other minor points and revisions:

- The α -Syn and A β structures selected to represent pathogenic fibrils are both *in vitro* generated structures that differ from solved *ex vivo* structures. Given that the authors are comparing functional versus pathogenic amyloids, they should report the comparisons using the known, pathogenic fibril structures for these proteins. It would be helpful to show more example structures in these comparative analyses, as the results section makes claims of hnRNPD_L-2 fibrils against all pathogenic fibrils.

Answer from authors: We agree with the reviewer that it would be interesting to compare the structure of pathological amyloids with functional amyloids. During the last two years, an important number of pathogenic *ex vivo* α -Syn and A β -42 structures have been elucidated. For this reason, we have included additional examples of tissue-derived α -Syn and A β -42 amyloid structures in our comparative analyses. These new structures have been incorporated to Supplementary Table 4, Extended Data Fig. 12,

and Extended Data Fig. 13 (see the new figures below). Revised versions of these two figures with the new comparisons have been included in the Extended Data file of the revised manuscript (pages 17 and 18) and a short comment incorporated in the main text (page 11).

Supplementary Table 4. Solvation free energy gain per molecule (ΔG) and per residue ($\Delta G/\text{residue}$) parameters calculated for hnRNPD-2, hnRNPA2, hnRNPA1, α -synuclein, hSAA1 and A β -42 amyloid fibrils.

	PDB accession code	ΔG (kcal/mol)	$\Delta G/\text{residue}$ (kcal/mol)
hnRNPA1	7BX7	-15.3	-0.34
hnRNPA2	6WQK	-20.7	-0.36
hnRNPD-2	7ZIR	-21.7	-0.43
α-synuclein*	6XYP	-49.3	-0.61
α-synuclein*	6XYQ	-50.0	-0.62
α-synuclein**	7NCA	-38.4	-0.63
α-synuclein**	7NCH	-43.2	-0.64
hSAA1*	6MST	-35.1	-0.65
α-synuclein**	7NCI	-44.4	-0.65
α-synuclein**	7NCG	-40.0	-0.66
α-synuclein*	6XYO	-53.3	-0.66
α-synuclein	6OSJ	-40.2	-0.66
α-synuclein**	7NCK	-45.5	-0.71
α-synuclein**	7NCJ	-43.4	-0.71
Aβ-42*	7Q4B	-26.5	-0.78
Aβ-42	5KK3	-21.9	-0.78
Aβ-42*	7Q4M	-26.5	-0.85

Note that proteins are ordered according to increasing $\Delta G/\text{residue}$ values. ΔG values were calculated with PDBePISA². *Fibrils obtained ex vivo. **Fibrils obtained upon seeding with ex vivo filaments.

Extended Data Fig. 12

Extended Data Fig. 12. hnRNPDL-2 fibrils display a highly polar surface interface. Surface representation of a) hnRNPDL-2 (PDB 7ZIR), b) human serum amyloid A (PDB 6MST), c) Amyloid β (A β)-42 (PDB 5KK3), d) α -synuclein (PDB 6OSJ), e) type 2 A β -42 (7Q4M) and f) type 3 α -synuclein (7NCK) fibrils cross-sections colored with the hydrophobicity levels of each residue. Note that a), c) and d) are amyloid fibrils obtained *in vitro*, while b) and e) constitute two examples of pathological amyloids obtained *ex vivo* from human tissues and f) is an amyloid fibril seeded by *ex vivo* filaments. hnRNPDL-2 has a unique strong hydrophobic spot visible in the core of the fibril. This region corresponds to the side chain of F231, which is located near the N-terminus. The hydrophobicity levels were assigned to each residue according to the Kyte-Doolittle scale⁷.

Extended Data Fig. 13

Extended Data Fig. 13. Stability of hnRNPDL-2 in comparison with other functional and pathological amyloid fibrils. Represented with fibril structures in two dimensions: solvation free energy gain (ΔG in Kcal/ml) per molecule (horizontal axis) and per residue (vertical axis). Amyloid fibril structures that are non-pathogenic and less stable are colored in green, whereas pathogenic and more stable amyloid filaments are colored in red. Note that the corresponding PDB codes are indicated below each structure.

- Lines 211-213, need correcting to read: “a value that is significantly lower in disease-associated fibrils...” instead of “significantly lower than in...”.

Answer from authors: We thank the reviewer for pointing out this error in the text. We have replaced “significantly lower” with “significantly higher” in the revised manuscript, page 10.

- Extended Data Fig. 9d, there’s a water H-bond that doesn’t appear to link to another visible atom?

Answer from authors: We thank the reviewer for pointing out this error in Extended Data Fig. 9d (Extended Data Fig. 11 in the revised version of the Extended Data file). In this figure, two of the H-bonds were misplaced. In fact, such interaction takes place with the adjacent water molecule, which is located at 2.8 Å from the N atom from the side chain of N241. This mistake has been corrected in the revised version of Extended Data Fig. 11 (page 16). To improve the clarity of the figure, we have also included the distances observed for all the H-bond interactions (see new version below).

Extended Data Fig. 11. Relevant interactions in the hnRNPDL-2 protofilament interface. d) Close-up view of the central water channel found in the hnRNPDL-2 amyloid core. Relevant hydrogen bonds established between water molecules in the channel and the adjacent residues from the first layer are depicted (yellow dashed lines). The corresponding distances are indicated in Å.

- Missing citation for Coot in the Materials and Methods.

Answer to the reviewer: We appreciate the reviewer’s suggestion. We have included the Coot citation in the revised Materials and Methods section.

- Please check over the figure legends, in several cases there is missing information e.g. what are the arrows pointing to in Fig. 1b, what is blue and what is green? What are the red dots in Fig. 2 (this only becomes clear once ED figures are looked at)?

Answer from authors: We thank the reviewer for the suggestion and for the accurate revision of our manuscript. We have carefully revised all the figure legends and improved figure descriptions. In particular, we added additional details on Fig. 1b legend. Fig. 1b and Extended Data Fig. 3 show representative confocal images of HeLa hnRNPD L KO cells transiently transfected with each of the different GFP-tagged hnRNPD L isoforms (DL-1, DL-2, and DL-3) or with the empty GFP vector (GFP). The images show the characteristic nuclear distribution of these proteins in green and the nuclear DNA, stained with Hoechst, in blue. This information has been incorporated to the revised Fig. 1b legend in the revised manuscript (page 6).

Moreover, we are sorry we did not mention that the red dots in Fig. 2 are water molecules. We have added this information to the Fig. 2 legend in the revised manuscript (page 8 and 9).

It would be helpful for consistent colours also to be used in Fig. 1a and ED Fig. 1.

Answer to the reviewer: According to the reviewer's suggestion, we have changed the colour code used in ED Fig. 1 to be consistent with the colours used in Fig. 1. The new version of this figure has been included in the Extended Data file (page 6).

- Could Fig. 1a be improved? Which residues correspond to exon 6 and how is the numbering reported for the different constructs with different N-terminal lengths.

Answer to the reviewer: We thank the reviewer for his/her suggestion, as residue numbering in Fig. 1a was somehow confusing. The three hnRNPD L isoforms have distinct numbering, which was not reflected in the first version of this figure. The residues shown for exon 6 correspond to the sequence comprised between Gly224 and Ser280 considering hnRNPD L isoform 2 numbering. We have improved Fig. 1a by providing the different isoform numbering, and we have also clarified that exon 6 corresponds to residues from 224 to 280 according to hnRNPD L-2 numbering. Please see the changes in the revised manuscript, pages 5 and 6.

Reviewer 3

REVIEWER COMMENTS

Reviewer #3 (Remarks to the Author):

Overview of relevant issues of the research:

This study brilliantly describes the full length fibrillar structure of hnRNPD L by cryo-electron-microscopy. This member of the ribonucleoprotein (RBNP) family shares a prion domain conferring aggregation attributes and nucleic acid binding domains relevant for biogenesis and metabolism of RNA.

Pathological mutations in several RNPs give rise to neurodegenerative diseases involving CNS, motor neuron, peripheral nerve and muscle that may appear as an isolated disorder (e.g. ALS) or as a multisystemic impairment.

Regarding the hnRNPD L disease-causing mutations only myopathic affection (LGMD-D3) has been reported. Pathologically, the affected cells display insoluble deposits and dislocation of the altered riboprotein usually accompanied by other related protein partners, along with signs of activation of diverse metabolic cell routes acting in defense or as expression of the degenerative cascade. Currently there are many gaps in the understanding of that complex scenario.

The study reveals that the hnRNPD L-2 isoform in its fibrillar conformation holds a particular structure different from other closely related RNPs (hnRNPA1, hnRNPA2), permitting to maintain the functional capabilities of nucleic acid processing while their prion amyloidogenic propensity and cytotoxic abilities are limited. An extensive physical entropic analysis of the peptide model gives a rational explanation for the behavior of this construct. The authors propose that a physiologic equilibrium between the hnRNPD L-2 monomeric and fibrillar conformation may be relevant for their connection with RNA metabolism, topologic distribution, and putative interaction with other proteins.

Importantly, analyses of the disease-related mutations yielded abnormal conformed peptide variants that similarly to the wt. lacked fibrillar aggregation propensity. This led the authors to postulate the existence of alternative patho-mechanisms to the prevailing thesis of insoluble phase transition / amyloidogenic deposition, which might include loss of function by a quick elimination of the mutated proteins via proteasome /autophagic routes or deleterious effects of the mutated species.

The authors appeal to the absence of protein aggregates in muscles from LGMD-3 patient's biopsies or mutated fly model supporting this hypothesis.

Other interesting contribution of this research deal with the description of the molecular structure and functional behavior of the three hnRNPD1 isoforms using a transfected cell model and in vitro protein analysis.

Comments and concerns:

This work significantly contributes to the understanding of the complex matter of ribonucleoprotein pathophysiology. The paper is well written, the experimental procedures clearly described, and figures and table of data are highly illustrative.

Answer from authors: We thank the reviewer for the positive comments on our work. In the following, we try to clarify all his/her concerns and incorporate suggestions.

However, there are some points worth mentioning:

1- Previous experiments indicates that hnRNPD1mutants exert a cell toxic effect (Navarro et al 2015). How can you interpret the discrepancy with the current results?

Answer from authors: The question makes perfect sense. The current study shows that the hnRNPD1-2 amyloid fibrils formed in vitro by the highly pure protein are not toxic for HeLa nor SH-SY5Y cell lines up to a concentration of 0.4 mg/ml, which is indeed exceedingly high for toxicity assays (Extended Data Fig. 8 in the revised file).

In the previous publication by Navarro et al.⁸, we addressed qualitatively the toxicity of the longest hnRNPD1-1 isoform in the form of Inclusion Bodies (IBs) produced in bacteria (*E. coli*). At that time, we were unaware of the relevance of the other isoforms generated by alternative splicing (hnRNPD1-2 & hnRNPD1-3) for this protein's function. IBs were used because we could not purify the soluble protein on that occasion.

hnRNPD1-1 IBs induced positive propidium iodide (PI) staining of the cells, suggesting they exerted toxicity. However, hnRNPD1-1 IBs were not fibrillar and displayed an amorphous morphology. In contrast to the fibrils described here, IBs expose hydrophobic regions that may interact and disrupt cellular membranes. Please be aware that we show now that when purified hnRNPD1-1 does not aggregate. Thus, the IBs we detected before likely result from the high expression levels attained in bacteria and probably were formed by nonspecific interactions between nascent polypeptide chains in the cytosol. We cannot also discard the presence of residual bacterial membrane components in this type of structure, such as lipopolysaccharides, that can contribute to their toxicity.

We believe that the nature of these particles, the different isoforms used in the two studies, and a certain heterogeneous composition of the IBs may explain the differences

in toxicity observed between both studies. Another possibility is that the presence of oligomeric species within IBs may account for the toxicity of hnRNPD1-1 IBs. In our new study, we evaluated the toxicity of mature amyloid fibrils, which are clearly innocuous, and we think they better reflect the assemblies formed by hnRNPD1-2 in the eukaryotic context.

2- Certainly, several LGMD-3 myopathologic studies coincide in the absence of sarcoplasmic protein aggregates (Viera et al 2014, Sun 2019, Berardo 2019). However, amyloid fibril / congophilic deposits were detected in some instances (Viera et al 2014, Vicente et al 2020). A remark on this issue would be of interest.

Answer from authors: We agree that a mention of this issue should be included. When we first made our fly model of LGMD3 myopathy, together with Paul J. Taylor, by overexpressing hnRNPD1-2 mutants⁹, we got very surprised not to find aggregates in myocytes. In the same flies, made in the same lab, overexpressed hnRNPA2/B1 and hnRNPA1 mutant proteins accumulated mainly in cytoplasmic inclusions and caused severe muscle degeneration, which also occurred in a mouse model expressing the mutant proteins¹⁰, and is consistent with what is observed in Inclusion Body Myopathies in which these two proteins are mutated.

The lack of an increased amyloid propensity for hnRNPD1-2 mutants in this article recapitulates the fly model and is consistent with the absence of sarcoplasmic protein aggregates in most patients. However, as the reviewer states, in some instances, congophilic inclusions have been identified, although the presence of hnRNPD1 in them has not yet been confirmed. In contrast, it has been shown that patients show diffuse hnRNPD1 in their muscles¹¹, which illustrates the histopathological complexity of this disease. In any case, we moderated our asseveration in the introduction (page 3) and have indicated the eventual detection of muscular inclusions in the revised text version (page 21).

3- In Viera's study TPNO-1 co-localized with hnRNPD1 in nucleus in normal control but the pattern changed in mutated giving a variable nuclear / perinuclear pattern. Though the issue is not addressed in this work, would it be of interest to make a quick remark on the potential implications of that interaction?

Answer from authors: This is an interesting aspect we briefly indicated in our previous text due to space limitations. "The accessibility of PY-NLS in both monomeric and fibrillar states of hnRNPD1-2 is consistent with the observation that, in humans, Kap β 2 co-localizes with both condensed and diffuse hnRNPD1 nuclear regions". Kap β 2 is also known as Transportin-1 or TNPO1.

According to Viera et al., the diffused pattern around the nuclei might also be observed in healthy people, but it is more frequently observed in patients than in normal muscles. This suggests that the transportation of hnRNPD2 by TNPO1 might be somehow affected by mutations¹². This possibility has now been included in the revised text (page 21 and 22). However, it is important to be cautious in this asseveration since, in Viera's study, it was not addressed the specific isoform that was miss-localized in patients.

4- Are there any hypothetical explanation why deleterious hnRNPD2 mutations are restricted to muscle?

Answer from authors: We thank the reviewer for this very interesting question. The reason why mutations in specific proteins have organ/tissue-specific effects is a subject of high interest, but in the majority of cases the scientific community does not have an answer.

In particular, pathogenic missense variants in RNA Binding Proteins cause a spectrum of diseases with pleomorphic phenotypic manifestations, including amyotrophic lateral sclerosis, motor neuron disease, frontotemporal dementia, inclusion body myopathy, distal myopathy, and Paget's disease of bone.

Also, for the same protein, different mutations might impact different tissues. For example, this is the case of hnRNPA2, with some mutations causing sporadic and familial motor neuron disease and others causing muscular dystrophy¹⁰.

5- I miss proposal delving into the alternative or concurrent patho-mechanisms implicated in LGMD-D3 to be undertaken on patient's muscle / cultivate cells, and disease models.

Answer from authors: Of course, this aspect is highly relevant, but we can only speculate here, because we still do not have a clear idea of hnRNPD2 function in human muscle. Because hnRNPD2 mutants are found in heterozygosity, they appear to act in a dominant manner. This observation raises the possibility of a loss of function (i.e., haploinsufficiency), a gain of toxic function, or a dominant-negative mechanism in which the mutated protein impacts the function of the WT form.

The data in this work, together with the previous one in the fly model, argue against a gain of toxicity caused by increased aggregation in the mutant forms. Indeed, even for hnRNPA2, the archetype of hnRNPs for which disease-associated mutations increase aggregation, disease-causing missense mutations that result in proteins that are more soluble than the WT protein, have recently been found. For these soluble variants, the problem seems to be reduced efficiency of nuclear import¹⁰.

This leads us to propose a loss of function mechanism; so far, the most robust evidence for this mechanism is the observation that hnRNPD is essential for muscle development in zebrafish, causing amyopathic phenotype when knocked down. Similarly, although hnRNPA1 mutations have been associated with a gain of toxic function upon aggregation, a recently generated hnRNPA1 knockout mice showed embryonic lethality because of muscle developmental defects in homozygosis. The study demonstrated that hnRNPA1 plays a critical and irreplaceable role in embryonic muscle development by regulating the expression and alternative splicing of muscle-related genes. Notably, a hnRNPA1^{+/-} heterozygotic mouse exhibited heart muscle defects¹³. We hypothesize that hnRNPD loss of function due to mutation might have a related effect, since emerging data suggest a role for LCD-containing RBPs in muscle regeneration, executing pre-mRNA splicing, stabilizing large muscle-specific transcripts, and aiding in their transport¹⁴. Ultimately, these reflections have been briefly introduced in the revised text discussion (page 22).

REFERENCES

1. Fraczkiewicz, R. & Braun, W. Exact and efficient analytical calculation of the accessible surface areas and their gradients for macromolecules. *Journal of Computational Chemistry* 19, 319-333 (1998).
2. Krissinel, E. & Henrick, K. Inference of macromolecular assemblies from crystalline state. *Journal of Molecular Biology* 372, 774-797 (2007).
3. Schrödinger, L. The PyMOL Molecular Graphics System. Vol. 2.0 edn.
4. Jurrus, E. et al. Improvements to the APBS biomolecular solvation software suite. *Protein Sci* 27, 112-128 (2018).
5. Kamei, D. & Yamada, M. Interactions of heterogeneous nuclear ribonucleoprotein D-like protein JKTBP and its domains with high-affinity binding sites. *Gene* 298, 49-57 (2002).
6. Gueroussov, S. et al. Regulatory Expansion in Mammals of Multivalent hnRNP Assemblies that Globally Control Alternative Splicing. *Cell* 170, 324-339 e23 (2017).
7. Kyte, J. & Doolittle, R.F. A simple method for displaying the hydrophobic character of a protein. *J Mol Biol* 157, 105-32 (1982).
8. Navarro, S., Marinelli, P., Diaz-Caballero, M. & Ventura, S. The prion-like RNA-processing protein HNRPD forms inherently toxic amyloid-like inclusion bodies in bacteria. *Microb Cell Fact* 14, 102 (2015).
9. Batlle, C. et al. hnRNPD Phase Separation Is Regulated by Alternative Splicing and Disease-Causing Mutations Accelerate Its Aggregation. *Cell Rep* 30, 1117-1128 e5 (2020).
10. Kim, H.J. et al. Mutations in prion-like domains in hnRNPA2B1 and hnRNPA1 cause multisystem proteinopathy and ALS. *Nature* 495, 467-73 (2013).
11. Vieira, N.M. et al. A defect in the RNA-processing protein HNRPD causes limb-girdle muscular dystrophy 1G (LGMD1G). *Hum Mol Genet* 23, 4103-10 (2014).

12. Kawamura, H. et al. Identification of the nucleocytoplasmic shuttling sequence of heterogeneous nuclear ribonucleoprotein D-like protein JKTBP and its interaction with mRNA. *J Biol Chem* 277, 2732-9 (2002).
13. Liu, T.Y. et al. Muscle developmental defects in heterogeneous nuclear Ribonucleoprotein A1 knockout mice. *Open Biol* 7(2017).
14. Vogler, T.O. et al. TDP-43 and RNA form amyloid-like myo-granules in regenerating muscle. *Nature* 563, 508-513 (2018).

REVIEWERS' COMMENTS

Reviewer #1 (Remarks to the Author):

The authors have satisfactorily addressed my comments

Reviewer #2 (Remarks to the Author):

This is an important and very interesting article that all three reviewers were impressed by. The authors have done an excellent job to address all of my comments and those of the other reviewers, with the outcome of an even better and clearer manuscript describing this fascinating story.

Reviewer #3 (Remarks to the Author):

Garcia-Pardo et al. make a comprehensive and straightforward rebuttal on my concerns about some apparent inconsistencies detected in their paper. Particularly the clarification why hnRNPD1-1 inclusions proved to be toxic in bacteria E Coli in a previous experiment while hnRNPD1-2 amyloid fibrils in the current one does not cause any harm to different cell models are highly convincing. Second, I applaud the tone down on stating the absence protein aggregates in patient's muscle biopsies because though hnRNPD1 deposits have not been documented, the presence of rimmed vacuoles, congofilic expression and tubule-filaments features occurring in a similar way to other IBM myopathies leave some gaps that requires future clarifications. I also appreciate their brief comments on the connection between hnRNPD1 and TNPO1 proteins useful to approach the understanding of uncovered pathogenic aspects of the muscle disorders triggered by mutations in their respective codifying genes.

In conclusion the clarifications and small modifications have strengthen the relate of the manuscript

Reviewer 1

REVIEWER COMMENTS

Reviewer #1 (Remarks to the Author):

The authors have satisfactorily addressed my comments

Reviewer 2

REVIEWER COMMENTS

Reviewer #2 (Remarks to the Author):

This is an important and very interesting article that all three reviewers were impressed by. The authors have done an excellent job to address all of my comments and those of the other reviewers, with the outcome of an even better and clearer manuscript describing this fascinating story.

Reviewer 3

REVIEWER COMMENTS

Reviewer #3 (Remarks to the Author):

Garcia-Pardo et al. make a comprehensive and straightforward rebuttal on my concerns about some apparent inconsistencies detected in their paper. Particularly the clarification why hnRNPD-1 inclusions proved to be toxic in bacteria E Coli in a previous experiment while hnRNPD-2 amyloid fibrils in the current one does not cause any harm to different cell models are highly convincing. Second, I applaud the tone down on stating the absence protein aggregates in patient's muscle biopsies because though hnRNPD deposits have not been documented, the presence of rimmed vacuoles, congofilic expression and tubule-filaments features occurring in a similar way to other IBM myopathies leave some gaps that requires future clarifications. I also appreciate their brief comments on the connection between hnRNPD and TNPO1 proteins useful to approach the understanding of uncovered pathogenic aspects of the muscle disorders triggered by mutations in their respective codifying genes.

In conclusion the clarifications and small modifications have strengthen the relate of the manuscript

Answer to the reviewers: We sincerely thank the reviewers for their very positive and thoughtful comments on our manuscript.